# Mini-Monkey: Alleviating the Semantic Sawtooth Effect for Lightweight MLLMs via Complementary Image Pyramid

**Mingxin Huang**[1], **Yuliang Liu**[2], **Dingkang Liang**[2], **Lianwen Jin**[*1,3,4], **Xiang Bai**[*2]

[1]South China University of Technology [2]Huazhong University of Science and Technology

[3] SCUT-Zhuhai Institute of Modern Industrial Innovation, Zhuhai, China

[4] INTSIG-SCUT Joint Lab on Document Analysis and Recognition

## ABSTRACT

Recently, scaling images to high resolution has received much attention in multi-modal large language models (MLLMs). Most existing practices adopt a sliding-window-style cropping strategy to adapt to resolution increase. Such a cropping strategy, however, can easily cut off objects and connected regions, which introduces semantic discontinuity and therefore impedes MLLMs from recognizing small or irregularly shaped objects or text, leading to a phenomenon we call the semantic sawtooth effect. This effect is particularly evident in lightweight MLLMs. To address this issue, we introduce a Complementary Image Pyramid (CIP), a simple, effective, and plug-and-play solution designed to mitigate semantic discontinuity during high-resolution image processing. In particular, CIP dynamically constructs an image pyramid to provide complementary semantic information for the cropping-based MLLMs, enabling them to richly acquire semantics at all levels. Furthermore, we introduce a Scale Compression Mechanism (SCM) to reduce the additional computational overhead by compressing the redundant visual tokens. Our experiments demonstrate that CIP can consistently enhance the performance across diverse architectures (e.g., MiniCPM-V-2, InternVL2, and LLaVA-OneVision), various model capacity (1B→8B), and different usage configurations (training-free and fine-tuning). Leveraging the proposed CIP and SCM, we introduce a lightweight MLLM, Mini-Monkey, which achieves remarkable performance in both general multimodal understanding and document understanding. On the OCRBench, the 2B-version Mini-Monkey even surpasses the 8B model InternVL2-8B by 12 score. Additionally, training Mini-Monkey is cheap, requiring only eight RTX 3090 GPUs. Code and models are available at https://github.com/Yuliang-Liu/Monkey.

## 1 INTRODUCTION

Recently, Large Language Models (LLMs) (Zhang et al., 2022; Brown et al., 2020; Touvron et al., 2023; OpenAI, 2023) have received significant attention for their robust text understanding and generation capabilities. Researchers are actively exploring ways to integrate vision encoders into LLMs to upgrade them to multimodal large language models (MLLMs) (Li et al., 2023b; Liu et al., 2023a; Bai et al., 2023). Some approaches employ a Q-former (Alayrac et al., 2022; Li et al., 2023b), while others (Liu et al., 2024d; Wang et al., 2023a) use linear projection. Despite the promising results, they are constrained to processing low-res images, which limits their ability to execute detailed scene analysis.

To address this limitation, much recent effect aims to enable MLLMs to process high-res images. One straightforward solution is to adopt a visual encoder that can tackle high-res images. However, developing a high-quality visual encoder demands substantial training resources (Bai et al., 2023; Chen et al., 2023). An alternative, more resource-efficient strategy is the non-overlapping crop-

---

[*]Corresponding authors.

This work was done when M. Huang was visiting Huazhong University of Science and Technology.

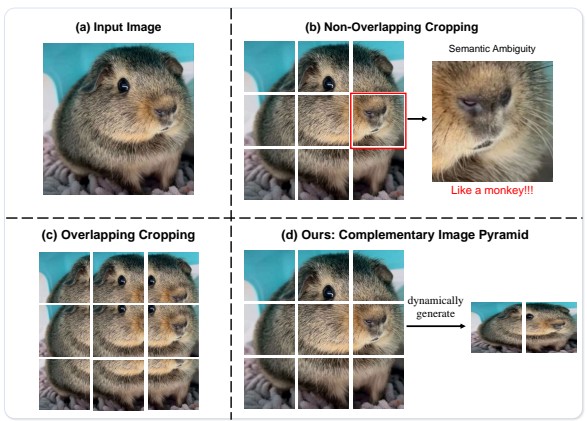

Figure 1: **Comparison of different image cropping strategies**. (a) Input image; (b) Non-overlapping cropping; (c) Overlapping cropping; (d) Ours: complementary image pyramid.

ping (Lin et al., 2023b; Liu et al., 2024c; Ye et al., 2023a; Li et al., 2024f; Chen et al., 2024b), which splits a high-res image into a set of low-res sub-images.

While the non-overlapping cropping strategy has shown promising results, it inevitably cuts off objects and connected regions, rendering difficulty for the MLLM in recognizing small or irregularly shaped objects due to semantic discontinuity, particularly in the context of document understanding. This mainly leads to two types of consequences: i) *semantic ambiguity*: if an object or character is divided, it may be misidentified; the nose of guinea pig in Fig. 1(b) looks much like a monkey after cropping, for instance; 2) *semantic damage*: if a word or sentence is segmented, the meanings of the segmented word will be changed completely; if the word 'breakdown' is divided into 'break' and 'down', the segmented words will have nothing to do with the original one (Liu et al., 2024f; Zhang et al., 2024). For simplicity, we call these phenomena the *semantic sawtooth effect* in this paper. To alleviate this effect, a rather straightforward idea is to adopt overlapping cropping. However, this strategy will result in the processing of much duplicate information, as presented in Fig. 1(c). This redundancy could potentially cause hallucinations in MLLMs. Moreover, it can even deteriorate performance according to our ablation studies in Sec. 4.3. Additionally, the semantic sawtooth effect can be observed more evidently in lightweight MLLMs. Larger MLLMs with enhanced comprehension capabilities and feature extraction capabilities often can alleviate this issue to some extent. Even when the object is segmented, these models can understand the objects through their powerful feature extraction.

To alleviate the semantic sawtooth effect more explicitly, we propose a plug-and-play approach, termed Complementary Image Pyramid (CIP). CIP can be easily integrated into a variety of cropping-based MLLMs, allowing them to tackle high-res images with reduced semantic sawtooth effect. CIP dynamically constructs an image pyramid that provides complementary semantic features for the MLLMs, enabling it rich acquire semantics at all levels. If object semantics are lost at one scale, they can be compensated by those from another scale. Different from previous work (Liu et al., 2024f; Huang et al., 2024) that addresses this issue by modifying the architecture of the model, our approach focuses on enriching the image semantics per se. Consequently, CIP can be easily integrated into a variety of MLLMs, allowing them to tackle high-res images with reduced semantic sawtooth effect. Considering that the CIP introduces some additional computational overheads, we further propose a Scale Compression Mechanism (SCM) for use in situations with limited computational resources. The SCM is both training-free and parameter-free. It leverages the well-trained attention layers of the LLM and the multi-scale information to generate attention weights, which in turn are used to compress redundant tokens. Utilizing the proposed CIP and SCM, we introduce a lightweight MLLM, Mini-Monkey.

Our experiments demonstrate the effectiveness of the proposed method: 1) 2B-parameter Mini-Monkey outperforms the InternVL2-2B by an average of $2.4\%$ across 17 benchmarks in terms of evaluation metrics; 2) Mini-Monkey achieves a score of 806 on the OCRBench, outperforming the 8B-parameter model InternVL2-8B by 12 score. Moreover, we observe that directly fine-tuning

well-performing pre-trained MLLM does not enhance, but rather degrades its performance. In contrast, fine-tuning with CIP can facilitate the training process to improve performance. In conclusion, the contributions of this work can be summarized as follows:

- CIP: a plug-and-play complementary image pyramid designed to alleviate the semantic sawtooth effect for multimodal large language models;
- Mini-Monkey: a lightweight, effective, and training-efficient multimodal large language model that integrates the complementary image pyramid and the scale compression mechanism;
- Our method achieves promising results on 8 general multimodal understanding benchmarks and 9 document understanding benchmarks, demonstrating the benefits of alleviating the semantic sawtooth effect.

## 2 Related Work

### 2.1 Multimodal Large Language Models

**Low-Resolution Input.** In recent years, Large Language Models (LLMs) have made significant progress (Zhang et al., 2022; Brown et al., 2020; Touvron et al., 2023; OpenAI, 2023). Drawing from this advancement, many efforts have been made to integrate a vision encoder into Large Language Models for vision-language understanding. A commonly employed approach is the linear projector method(Liu et al., 2024d; Wang et al., 2023a), which maps the output of the vision encoder to the same feature space as the text features of the Large Language Models. Some methods, such as Q-Former (Li et al., 2023b), Perceiver Resampler (Alayrac et al., 2022), or Abstractor (Ye et al., 2023b), introduce a set of learnable queries to facilitate this integration. Despite these notable advances, previous methods often struggle with detailed scene understanding due to limitations in resolution.

**Naïve High-Resolution Strategy vs. Cropping Strategy** To address this issue, recent research has adopted two primary strategies: 1) Naïve High-Resolution Strategy. This Strategy leverages the vanilla ViT to handle images of any resolution and aspect ratio (Wang et al., 2024; Liu et al., 2024g). However, these methods require additional training data and parameters, or processing attention over high-res images significantly increases computational demands. 2) An alternative, more resource-efficient method is the cropping strategy, which divides the high-res image into multiple lower-resolution sub-images for processing (Li et al., 2024f; Lin et al., 2023b; Ye et al., 2023a; Chen et al., 2024b; Dubey et al., 2024). At a resolution of 2240x2240, the Naïve High-Resolution Strategy requires around 50GB of GPU memory, whereas the cropping strategy needs only about 16GB. These results demonstrate the cropping strategy's efficiency in managing high-resolution images, making it a more practical choice for resource-constrained environments.

Although the cropping strategy achieves promising results on several multimodal benchmarks, it will inevitably result in a semantic sawtooth effect: 1) If an object or character is divided, it may not be recognized; 2) If the word or sentence is segmented, the semantic damage of the segmented word will be caused. For example, the word 'Breakdown' may be divided into 'Break' and 'down', causing semantic damage to the segmented word. This will limit the model's ability to understand the detailed scene. Although some methods (Liu et al., 2024f; Huang et al., 2024) attempt to address this issue by introducing new modules, they introduce additional parameters to the original model and require training this module from scratch. In contrast, the proposed CIP is designed to be seamlessly integrated without introducing additional parameters, offering a plug-and-play solution.

### 2.2 Lightweight Multimodal Large Language Models

Due to the substantial computational costs associated with multimodal large language models (MLLMs), some recent efforts have focused on developing more efficient models for rapid development and real-world applications. For instance, LLaVA-Phi (Zhu et al., 2024) and Imp (Shao et al., 2024) integrate a lightweight large language model with a vision encoder to develop a powerful multimodal system. MobileVLM (Chu et al., 2023) further conserves resources by integrating a lightweight downsampling projector that reduces the number of visual tokens. Bunny (He et al., 2024) advances efficiency through an effective data compression technique, which minimizes the

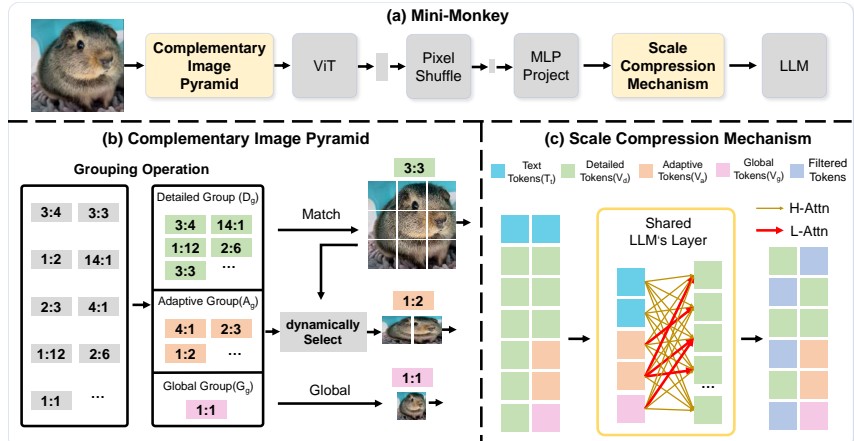

Figure 2: **Overall architecture of Mini-Monkey**. H-Attn represents high attention weights. L-Attn represents low attention weights. The tokens with low attention weights will be filtered. The shared LLM's Layer represents using the block layer from LLM in SCM. The pixel shuffle operation is utilized to reduce the number of visual tokens to one-quarter of the original.

required pretraining dataset. TinyGPT-V (Yuan et al., 2023) adopts a multi-stage training process specifically designed for lightweight multimodal models. The above models support only low-res input. To improve detailed scene understanding, one of the most commonly used methods is the cropping strategy. For instance, InternVL2-2B (Chen et al., 2024b) enhances the performance of lightweight MLLMs by adopting a dynamic high-res cropping strategy. Despite these advancements, the cropping strategy will introduce a semantic sawtooth effect, which significantly limits the performance of lightweight multimodal large language models. Larger MLLMs with enhanced comprehension capabilities can alleviate this issue to some extent, as discussed in Sec. 4.4.

## 3 MINI-MONKEY

The overall architecture is illustrated in Fig. 2. Mini-Monkey consists of a CIP, a vision encoder, an MLP layer, a Scale Compression Mechanism, and a Large Language Model (LLM). Initially, CIP dynamically generates an image pyramid based on the resolution of input images. Then, we divide these images into a set of sub-images. These sub-images are then processed by the vision encoder and MLP layer to extract image tokens. The Scale Compression Mechanism adjusts these image tokens and forwards them to the LLM, which subsequently generates the final answers.

### 3.1 COMPLEMENTARY IMAGE PYRAMID

Existing cropping strategy (Li et al., 2024f; Chen et al., 2024b) directly divides the high-res images into a set of sub-images that will lead to a semantic sawtooth effect. To address this issue, we propose a plug-and-play method, termed complementary image pyramid (CIP), to promote synergy among images at varying scales to alleviate the semantic sawtooth effect. The process of CIP is shown in Fig. 2 (b).

**Grouping Operation.** We begin by generating a set of pre-defined aspect ratios, which we define as follows: $\{g = (n_h \times n_w) | N_{min} \le n_h \cdot n_w \le N_{max}, n_h \in \mathbb{N}, n_w \in \mathbb{N}\}$. The $n_h$ and $n_w$ represent the number of height and width of the grid $g$. $N_{max}$ is the maximum number of tiles. $N_{min}$ is the minimum number of tiles. These aspect ratios are then categorized into three groups through a grouping operation, including a detailed group $D_g$, an adaptive group $A_g$, and a global group $G_g$. The classification is based on the following criteria: (1) Aspect ratios that are between $\frac{1}{3} * N_{max}$ and $N_{max}$ tiles being allocated to the detailed group, enabling the largest possible image size and thus a clearer depiction of the objects within. (2) For aspect ratios producing between $\frac{1}{8} * N_{max}$ and $\frac{1}{3} * N_{max}$ tiles, we classify them into the adaptive group, which is responsible for enhancing the fine details at the borders of the crops. (3) The 1:1 aspect ratio is designated to the global group,

providing a low-res, comprehensive view of the whole image. The grouping operation generates three groups of different aspect ratios for generating the image.

**Dynamically Generating Images.** After the grouping operation, we will generate three images from each group. First, we calculate the absolute differences between the aspect ratio of the input image and the aspect ratios within the detailed group. Then, the ratio that has the smallest absolute difference from the input image's aspect ratio is selected as the matched ratio, denoted as $D_h, D_w$. $D_h$ is the number of height tiles and $D_w$ is the number of width tiles. Once a matched aspect ratio is determined, the image is resized to the corresponding resolution. For example, a 1288×1257 image would be resized to 1344×1344. The resized image is then divided into tiles of 448×448 pixels. After obtaining the detailed image, the adaptive group will dynamically generate an aspect ratio based on $D_h, D_w$, ensuring that the cropping lines on the detailed group and those on the adaptive group do not overlap. First, we eliminate any aspect ratios in the adaptive group that are exact multiples of $D_h, D_w$. When the $A_h$ is 1, it means that there is no required cropping. Therefore, they can be integer multiples in such cases. This process can be formulated as follows:

$$\forall k \in \mathbb{Z}, \forall i \in \{h, w\}, \begin{cases} D_i = k \cdot A_i, & \text{if } A_i = 1, \\ D_i \neq k \cdot A_i, & \text{otherwise.} \end{cases} \tag{1}$$

where $A_h$ and $A_w$ denote the height and width components of the aspect ratios in the adaptive group, respectively. Then, we will resize the image by selecting the ratio closest to the aspect ratio of the original image from the remaining aspect ratio.

Because the vision encoder processes each tile independently, existing cropping-based MLLMs fail to capture feature interactions between different tiles. In our method, the adaptive group employs a distinct aspect ratio to partitioning windows compared to the detailed group, thereby simulating cross-tile interaction features and providing the cropping positions information for the detailed group. Similarly, the global group provides the cross-tile interaction features and the cropping positions information for the adaptive component. Three groups provide complementary semantic information and multi-scale information for the model, enabling the model to better capture finer details and handle objects of different sizes in images. Different from the previous method (Liu et al., 2024f; Huang et al., 2024), the proposed CIP alleviates the semantic sawtooth effect from the perspective of the image, bringing several advantages: (i) it is plug-and-play, requiring no additional parameters; (ii) it seamlessly integrates with existing MLLMs that utilize cropping strategies, leading to consistent performance improvements; and (iii) it can be utilized without training and its effectiveness can be further improved through fine-tuning.

## 3.2 Scale Compression Mechanism

Although the proposed CIP significantly enhances model performance, certain scenarios may restrict the level of computational resources available. To tackle this challenge, we introduce a parameter-free token compression method called the Scale Compression Mechanism (SCM), which is used to reduce the visual tokens, as shown in Fig. 2 (c). The detailed group provides tokens with lower information density, whereas the adaptive and global groups yield tokens that are more information-dense. Therefore, we primarily focus on compressing the tokens from the detailed group. Previous work demonstrates that a well-trained LLM from MLLM can effectively select the necessary visual features based on the input question (Chen et al., 2024a). Consequently, SCM utilizes the layers of the LLM from a well-trained MLLM to select visual tokens without generating any additional parameters. The input visual token including $\mathbf{V_d} \in \mathbb{R}^{L_1 \times C}$, $\mathbf{V_a} \in \mathbb{R}^{L_2 \times C}$, and $\mathbf{V_g} \in \mathbb{R}^{L_3 \times C}$, and the textual token $\mathbf{T_t} \in \mathbb{R}^{T \times C}$ will be sent into an LLM's Layer. $\mathbf{V_d}$ represents the tokens from the detailed group. $\mathbf{V_a}$ represents the tokens from the adaptive group. $\mathbf{V_g}$ represents the tokens from the global group. We utilize the first and second layers of LLM to compress the tokens. The LLM's Layer will output an attention map. We use the visual token from the adaptive group, global group, and textual token to attend to the visual token from the detailed group. The calculation of the attention can be formulated as follows:

$$\mathbf{Q} = \texttt{cat}(\mathbf{V_a}, \mathbf{V_g}, \mathbf{T_t}), \tag{2}$$

$$\mathbf{Attn_w} = \texttt{softmax}(\frac{(\mathbf{Q} + \text{PE}(\mathbf{Q}))(\mathbf{V_d} + \text{PE}(\mathbf{V_d}))^T}{\sqrt{D}}). \tag{3}$$

Table 1: Comparison with SoTA models on 8 multimodal benchmarks. General multimodal benchmarks encompass: MME (Fu et al., 2023), RealWorldQA (X.ai, 2024), AI2D test (Kembhavi et al., 2016), CCBench (Liu et al., 2023b), SEED Image (Li et al., 2023a), HallusionBench (Guan et al., 2023), and POPE (Li et al., 2023c). Additionally, the math dataset includes MathVista testmini (Lu et al., 2023). The MME results we report are the sum of the perception and cognition scores. [§] represents the results from the OpenCompass leaderboard (Contributors, 2023).

| model | #param | General Multimodal Benchmarks | | | | | | | Math |
| | | MME | RWQA | AI2D | CCB | SEED | HallB | POPE | MathVista |
| --- | --- | --- | --- | --- | --- | --- | --- | --- | --- |
| QWEN-VL (Bai et al., 2023) | 7B | 1848.3 | 49.3[§] | 63[§] | 65.7[§] | 52.5[§] | 29.9[§] | 70[§] | 34.9[§] |
| Mini-Gemini (Li et al., 2024e) | 35B | 2141.0 | – | – | – | – | – | – | 43.3 |
| LLaVA-NeXT (Liu et al., 2024c) | 35B | 2028.0 | – | 74.9 | 49.2 | 75.9 | 34.8 | 89.6[§] | 46.5 |
| InternVL 1.2 (Chen et al., 2024c) | 40B | 2175.4 | 67.5 | 79.0 | 59.2 | 75.6 | 47.6 | 88.0 | 47.7 |
| InternVL 1.5 (Chen et al., 2024b) | 26B | 2187.8 | 66.0 | 80.7 | 69.8 | 76.0 | 49.3 | 88.3 | 53.5 |
| DeepSeek-VL (Lu et al., 2024) | 1.7B | 1531.6 | 49.7[§] | 51.5[§] | 37.6[§] | 43.7[§] | 27.6[§] | 85.9[§] | 29.4 |
| Mini-Gemini (Li et al., 2024e) | 2.2B | 1653.0 | - | - | - | - | - | - | 29.4 |
| Bunny-StableLM-2 (He et al., 2024) | 2B | 1602.9 | - | - | - | 58.8 | - | 85.9 | - |
| MiniCPM-V-2 (Yao et al., 2024) | 2.8B | 1808.6 | 55.8[§] | 62.9[§] | 48.0[§] | - | 36.1[§] | 86.3[§] | 38.7 |
| InternVL 2 (Chen et al., 2024b) | 2B | 1876.8 | 57.3 | 74.1 | 74.7 | 70.9[§] | 37.9 | 85.2[§] | 46.3 |
| Mini-Monkey (ours) | 2B | 1884.2 | 57.9 | 74.8 | 75.5 | 71.3 | 38.8 | 88.0 | 47.3 |

where PE represents the position encoding and $D$ denotes the dimension of the LLM. $Cat()$ represents the sequence concatenation operation. After computing the attention mechanism, we average the first dimension of the attention map $\mathbf{Attn_w} \in \mathbb{R}^{(L_2+L_3+T) \times L_1}$ to obtain a weight vector $\mathbf{W_a} \in \mathbb{R}^{L_1}$. Subsequently, we select the top $K$ visual features from detailed layers based on this weight vector $\mathbf{W_a}$. These selected tokens, along with tokens from the adaptive group, global group, and textual token, are input into the LLM to generate the results. Compared to FastV (Chen et al., 2024a), SCM works in conjunction with the CIP and is more targeted by using tokens with high relative information density to compress tokens with low information density.

## 4 EXPERIMENTS

### 4.1 IMPLEMENTATION DETAILS

We use InternVL2-2B (Chen et al., 2024b) as the Baseline to develop the Mini-Monkey. Following previous work (Chen et al., 2024c), we use the (448, 448) as the input resolution of InternViT. The training datasets used to train the model include DocVQA (Mathew et al., 2021), ChartQA (Masry et al., 2022), DVQA (Kafle et al., 2018), AI2D (Kembhavi et al., 2016), GeoQA+ (Cao & Xiao, 2022), and LLaVA-150K (zh) (Liu et al., 2024d). We use the AdamW (Loshchilov & Hutter, 2017) as the optimizer. The base learning rate is 4e-8. We limit the maximum number $N_{max}$ to 24 and the minimum number $N_{min}$ is 1.

**Evaluation.** Following the previous work (He et al., 2024; Chen et al., 2024b), we evaluate Mini-Monkey on eleven general multimodal understanding benchmarks, including MathVista testmini (Lu et al., 2023), SEED Image (Li et al., 2023a), RealWorldQA (X.ai, 2024), AI2D test (Kembhavi et al., 2016), POPE (Li et al., 2023c), CCBench (Liu et al., 2023b), MME (Fu et al., 2023), and HallusionBench (Guan et al., 2023). For document understanding, following the previous work (Liu et al., 2024f), we employ two distinct types of metrics to verify the performance of Mini-Monkey. Initially, we leverage the standard metrics provided by the benchmarks to evaluate Mini-Monkey. We utilize benchmarks such as ChartQA (Masry et al., 2022), DocVQA (Mathew et al., 2021), InfoVQA (Mathew et al., 2022), TextVQA (Singh et al., 2019), STVQA (Biten et al., 2019), FUNSD (Jaume et al., 2019), SROIE (Huang et al., 2019), POIE (Kuang et al., 2023) and OCR-Bench (Liu et al., 2023c). We also apply the accuracy metric to verify the performance. Further details on this metric and the used benchmarks can be referenced in appendix A.8.

### 4.2 COMPARISON WITH STATE-OF-THE-ART METHODS

**General Multimodal Understanding.** We evaluate Mini-Monkey on general multimodal understanding following (He et al., 2024; Chen et al., 2024b). The results are shown in Tab. 1. Mini-Monkey surpasses other 2B-parameter models on 8 benchmarks. The results indicate that CIP enhances Mini-Monkey's perception ability, thereby improving its capability to handle general multi-

Table 2: Comparison to state-of-the-art MLLMs on OCR-related Tasks. Mini-Monkey achieves the best results among the 2B-parameter MLLMs. $\S$ represents the results from the OpenCompass leaderboard (Contributors, 2023).

| Model | Model Size | DocVQA$^{Test}$ | ChartQA$^{Test}$ | InfoVQA$^{Test}$ | TextVQA$^{Val}$ | OCRBench |
|---|---|---|---|---|---|---|
| TextMonkey (Liu et al., 2024f) | 9B | 73.0 | 66.9 | 28.6 | 65.6 | 558 |
| TextHawk (Yu et al., 2024) | 7B | 76.4 | 66.6 | 50.6 | — | — |
| DocKylin (Zhang et al., 2024) | 7B | 77.3 | 46.6 | 66.8 | — | — |
| HiRes-LLaVA (Huang et al., 2024) | 7B | 74.7 | 61.5 | 48.0 | 65.4 | — |
| LLaVA-UHD (Xu et al., 2024) | 13B | — | — | — | 67.7 | — |
| CogAgent (Hong et al., 2024) | 17B | 81.6 | 68.4 | 44.5 | 76.1 | 590 |
| UReader (Ye et al., 2023a) | 7B | 65.4 | 59.3 | 42.2 | 57.6 | — |
| DocOwl 1.5 (Hu et al., 2024a) | 8B | 82.2 | 70.2 | 50.7 | 68.6 | — |
| HRVDA (Liu et al., 2024a) | 7B | 72.1 | 67.6 | 43.5 | — | — |
| TextSquare (Tang et al., 2024) | 7B | 84.3 | 79.4 | 51.5 | 66.8 | 622 |
| IXC2-4KHD (Dong et al., 2024b) | 8B | 90.0 | 81.0 | 68.6 | 77.2 | 675 |
| InternVL 1.5 (Chen et al., 2024b) | 26B | 90.9 | **83.8** | 72.5 | **80.6** | 724 |
| InternVL 2 (Chen et al., 2024b) | 8B | **91.6** | 83.3 | **74.8** | 77.4 | **794** |
| GLM4-V (GLM et al., 2024) | 9B | - | - | - | - | 786 |
| Vary-toy (Wei et al., 2024) | 1.8B | 65.6 | 59.1 | - | - | - |
| MiniCPM-V 2.0 (Yao et al., 2024) | 2.8B | 71.9 | 55.6$^\S$ | - | 74.1 | 605 |
| InternVL 2 (Chen et al., 2024b) | 2B | 86.9 | 76.2 | 58.9 | 73.4 | 784 |
| Mini-Monkey (Ours) | 2B | **87.4** | **76.5** | **60.1** | **76.0** | **806** |

Table 3: Quantitative accuracy (%) comparison of our model with existing multimodal large language models (MLLMs) on several benchmarks. Following TextMonkey (Liu et al., 2024f), we use the accuracy metrics to evaluate our method.

| Method | Scene Text-Centric VQA | | Document-Oriented VQA | | | KIE | | |
|---|---|---|---|---|---|---|---|---|
| | STVQA | TextVQA | DocVQA | InfoVQA | ChartQA | FUNSD | SROIE | POIE |
| BLIP2-OPT-6.7B (Li et al., 2023b) | 20.9 | 23.5 | 3.2 | 11.3 | 3.4 | 0.2 | 0.1 | 0.3 |
| mPLUG-Owl (Ye et al., 2023b) | 30.5 | 34.0 | 7.4 | 20.0 | 7.9 | 0.5 | 1.7 | 2.5 |
| InstructBLIP (Dai et al., 2023) | 27.4 | 29.1 | 4.5 | 16.4 | 5.3 | 0.2 | 0.6 | 1.0 |
| LLaVAR (Zhang et al., 2023) | 39.2 | 41.8 | 12.3 | 16.5 | 12.2 | 0.5 | 5.2 | 5.9 |
| BLIVA (Hu et al., 2024b) | 32.1 | 33.3 | 5.8 | 23.6 | 8.7 | 0.2 | 0.7 | 2.1 |
| mPLUG-Owl2-8 (Ye et al., 2024) | 49.8 | 53.9 | 17.9 | 18.9 | 19.4 | 1.4 | 3.2 | 9.9 |
| LLaVA1.5-7B (Liu et al., 2024b) | 38.1 | 38.7 | 8.5 | 14.7 | 9.3 | 0.2 | 1.7 | 2.5 |
| TGDoc (Wang et al., 2023b) | 36.3 | 46.2 | 9.0 | 12.8 | 12.7 | 1.4 | 3.0 | 22.2 |
| UniDoc (Feng et al., 2023b) | 35.2 | 46.2 | 7.7 | 14.7 | 10.9 | 1.0 | 2.9 | 5.1 |
| DocPedia (Feng et al., 2023a) | 45.5 | 60.2 | 47.1 | 15.2 | 46.9 | 29.9 | 21.4 | 39.9 |
| Monkey-8B (Li et al., 2024f) | 54.7 | 64.3 | 50.1 | 25.8 | 54.0 | 24.1 | 41.9 | 19.9 |
| InternVL-8B (Chen et al., 2024c) | 62.2 | 59.8 | 28.7 | 23.6 | 45.6 | 6.5 | 26.4 | 25.9 |
| InternLM-XComposer2-7B (Dong et al., 2024a) | 59.6 | 62.2 | 39.7 | 28.6 | 51.6 | 15.3 | 34.2 | 49.3 |
| TextMonkey-9B (Liu et al., 2024f) | 61.8 | 65.9 | 64.3 | 28.2 | 58.2 | 32.3 | 47.0 | 27.9 |
| InternVL2-2B (Chen et al., 2024b) | 65.6 | 66.2 | 76.7 | 46.8 | **67.6** | 42.0 | 68.0 | 66.8 |
| Mini-Monkey-2B (Ours) | **67.2** | **68.8** | **78.4** | **50.0** | 67.3 | **43.2** | **70.5** | **71.2** |

modal understanding tasks. Additionally, on the POPE benchmark, which evaluates hallucinations in MLLMs, Mini-Monkey outperforms the Baseline InternVL2-2B by 2.8%, demonstrating that CIP can also mitigate hallucinations in MLLMs.

**Document Understanding.** For the first type of metric, the results are presented in Tab. 2. Compared to Baseline InternVL2-2B, our method outperforms it by 2.6%, 1.2%, and 22 for TextVQA, InfoVQA, and OCRBench, respectively. The CIP provides the model with complementary semantic and multi-scale information, enhancing its ability to perceive fine-grained and varying-sized text. Due to the small original resolution of ChartQA, it is less impacted by cropping operations, resulting in a minor improvement from our method. With these complementary semantic and multi-scale information, on the OCRBench, Mini-Monkey even surpasses the 8B-parameter Large Multimodal Model InternVL2-8B and the 9B-parameter Large Multimodal Model GLM4-V by 12 and 20, respectively. For the accuracy metric, the results are shown in Tab. 3. Mini-Monkey outperforms the InternVL2-2B by 2.6%, 3.2%, and 4.4% on TextVQA, InfoVQA, and POIE, respectively. OCR-related tasks are utilized to evaluate the fine-grained recognition capabilities of the MLLM. The results from these tasks demonstrate the effectiveness of CIP in enhancing such capabilities.

## 4.3 ABLATION STUDY

In this section, we perform ablation studies on both general multimodal understanding and document understanding benchmarks to validate the effectiveness of our method. We adopt the TextVQA (Singh et al., 2019), OCRBench (Liu et al., 2023c), HallusionBench (Guan et al., 2023), MME (Fu et al., 2023), and POPE (Li et al., 2023c) to conduct ablation studies.

**Complementary Image Pyramid.** We conducted ablation studies to investigate the effectiveness of the CIP. We compared our method with several alternatives: The dynamic high-res strategy (Chen et al., 2024b), which maintains aspect ratios to increase resolution. The fixed-size high-res strategy (Li et al., 2024f), which uses a fixed size to increase resolution. The overlapping cropping

Table 4: Ablation study of Complementary Image Pyramid. We compare our method with the existing cropping strategy and the overlay cropping strategy.

| Model | Resolution Strategy | TextVQA | OCRBench | MME | HallB | POPE | Flops(B) | Latency/Example |
|---|---|---|---|---|---|---|---|---|
| Baseline | Dynamic high-res Strategy (Chen et al., 2024b) | 73.4 | 784 | 1876.8 | 37.9 | 85.2 | **349.4** | **1.0s** |
| Baseline | Fixed Size high-res Strategy (Li et al., 2024f) | 74.2 | 772 | 1824.5 | 37.6 | 85.0 | 510.9 | 1.1s |
| Baseline | Overlapping Cropping Strategy | 70.6 | 758 | 1874.1 | 36.8 | 83.5 | 393.1 | 1.1s |
| Baseline | Multi-Scale Strategy (Shi et al., 2024) | 74.8 | 776 | 1846.8 | 38.1 | 85.3 | 559.2 | 1.6s |
| Mini-Monkey (Ours) | Complementary Image Pyramid | **76.0** | **806** | **1884.2** | **38.8** | **88.0** | 531.3 | 1.3s |

Table 5: Ablation study of the impact of different components in CIP.

| | Model | Detailed Component | Global Component | Adaptive Component | TextVQA | OCRBench | MME | HallB | POPE |
|---|---|---|---|---|---|---|---|---|---|
| r1 | InternVL2-2B | | ✓ | | 62.5 | 385 | 1686.2 | 34.8 | 81.8 |
| r2 | InternVL2-2B | | ✓ | ✓ | 70.5 | 473 | 1806.1 | 37.4 | 86.0 |
| r3 | InternVL2-2B | ✓ | | | 60.8 | 624 | 1842.3 | 37.4 | 85.3 |
| r4 | InternVL2-2B | ✓ | | ✓ | 74.8 | 782 | 1874.2 | **39.0** | 87.5 |
| r5 | InternVL2-2B | ✓ | ✓ | | 74.6 | 785 | 1853.5 | 37.6 | 87.6 |
| r6 | InternVL2-2B | ✓ | ✓ | ✓ | **76.0** | **806** | **1884.2** | 38.8 | **88.0** |

Table 6: Ablation study of the different compression rates of SCM.

| Compression Rate | 0.0 | 0.1 | 0.2 | 0.3 | 0.4 | 0.5 | 0.7 | 0.9 |
|---|---|---|---|---|---|---|---|---|
| MME | 1884.2 | 1884.7 | 1879.8 | 1878.5 | 1876.3 | 1886.0 | 1871.7 | 1870.2 |
| Flops (B) | 446.9 | 414.9 | 383.6 | 353.0 | 323.0 | 293.7 | 237.0 | 171.4 |
| Latency/Example | 0.83s | 0.78s | 0.73s | 0.67s | 0.63s | 0.59s | 0.51s | 0.49s |

strategy uses a high-res approach but crops with overlay. The multi-scale strategy (Shi et al., 2024), which introduces a multi-scale strategy to the MLLM. As presented in Tab. 4, the proposed CIP achieved the best results on both general multimodal understanding and document understanding without significantly increasing latency or computational load. The over-overlay cropping strategy, instead of improving the model's performance, actually degraded it.

**Various Model Capacity.** We performed ablation studies to assess the impact of CIP on models with varying model capacities. As illustrated in Table 8, CIP consistently improves the performance of varying model capacities, illustrating the general applicability of our approach.

**Different Usage Configurations.** To further validate the improvements introduced by CIP, we performed experiments on usage configurations: a training-free configuration and a fine-tuning configuration. As shown in Tab. 7, CIP demonstrates improvements in performance even when applied without training. The performance can be further improved with fine-tuning. Additionally, we surprisingly find that CIP can even facilitate the model fine-tuning process. As presented in the second line in Tab. 7, direct fine-tuning of the Baseline model not only failed to improve performance but, in some cases, led to a decline. Conversely, incorporating CIP during the fine-tuning of the Baseline resulted in substantial improvements in both general multimodal understanding and document understanding, as evidenced in the fourth line of Tab. 7.

**Incorporating CIP to various MLLMs.** The proposed CIP can be seamlessly integrated into crop-based MLLMs. To demonstrate its effectiveness, we incorporated CIP into various structures of MLLM, such as MiniCPM-V-2 (Yao et al., 2024), InternVL 2 (Chen et al., 2024b), LLaVA-OV (Li et al., 2024b). The results shown in Tab. 8 show that ClP can be seamlessly integrated into various MLLMs and consistently improves their performance.

**The Number of Sub-Images.** To investigate whether the performance enhancement is attributed to an increase in the number of sub-images, we performed an experiment by incrementally raising the sub-image count for the Baseline. The findings, summarized in Tab.10, indicate that increasing the number of sub-images does not lead to better performance; instead, it may result in a decline. In contrast, CIP can effectively improve the performance of the model, demonstrating its effectiveness.

**The impact of different components in CIP.** To evaluate the importance of each component in the CIP, we performed ablation studies using the InternVL2-2B. As presented in Tab. 5, the results indicate that using only the global component or only the detailed component results in a significant performance drop, as shown in r1 and r3 of Tab. 5. By comparing the r1 and r2, as well as r3 and r4, in Tab. 5, we find that adding an Adaptive component significantly improves the performance. When using both the detailed component and the global component, adding the adaptive component leads to further improvements, as shown in r5 and r6 of Tab. 5. Furthermore, the removal of any one of the

Table 7: Exploring different usage configurations of CIP. Train represents fine-tuning the model.

| Model | CIP | Train | TextVQA | OCRBench | MME | HallB | POPE |
|---|---|---|---|---|---|---|---|
| InternVL2-2B | × | × | 73.4 | 784 | 1876.8 | 37.9 | 85.2 |
| InternVL2-2B | × | ✓ | 73.3 (-0.1) | 787 (+3) | 1858.3 (-18.5) | 37.3 (-0.6) | 85.3 (+0.1) |
| InternVL2-2B | ✓ | × | 75.2 (+1.8) | 800 (+17) | 1881.9 (+5.1) | 38.7 (+0.8) | 86.7 (+1.5) |
| InternVL2-2B | ✓ | ✓ | 76.0 (+2.6) | 806 (+22) | 1884.2 (+7.4) | 38.8 (+0.9) | 88.0 (+2.8) |

Table 8: Ablation study of incorporating complementary image pyramid (CIP) to other MLLMs. [§] represents the results from the OpenCompass leaderboard (Contributors, 2023).

| Model | CIP | TextVQA | OCRBench | MME | HallB | POPE |
|---|---|---|---|---|---|---|
| MiniCPM-V-2-2.8B | × | 74.1 | 605 | 1808.6 | 36.1[§] | 86.3[§] |
| MiniCPM-V-2-2.8B | ✓ | 76.0 (+1.9) | 627 (+22) | 1819.5 (+10.9) | 36.5 (+0.4) | 87.1 (+0.8) |
| LLaVA-OV-0.5B | × | 65.3 | 577 | 1478.0 | 28.1 | 86.7 |
| LLaVA-OV-0.5B | ✓ | 66.2 (+0.9) | 600 (+23) | 1482.6 (+4.6) | 28.8 (+0.7) | 87.7 (+1.0) |
| InternVL2-1B | × | 70.5 | 754 | 1794.4 | 33.4 | 84.9[§] |
| InternVL2-1B | ✓ | 72.3 (+1.8) | 772 (+18) | 1801.5 (+7.1) | 34.3 (+0.9) | 85.7 (+0.8) |
| InternVL2-2B | × | 73.4 | 784 | 1876.8 | 37.9 | 85.2 |
| InternVL2-2B | ✓ | 76.0 (+2.6) | 806 (+22) | 1884.2 (+7.4) | 38.8 (+0.9) | 88.0 (+2.8) |
| InternVL2-8B | × | 77.4 | 794 | 2210.3 | 45.0[§] | 84.2[§] |
| InternVL2-8B | ✓ | 79.3 (+1.9) | 809 (+15) | 2226.4 (+16.1) | 45.4 (+0.4) | 84.8 (+0.6) |

Table 9: Ablation study of the scale compression mechanism. We used different compression ratios to compare with FastV (Chen et al., 2024a). (0.5) represents 50% compression and (0.9) represents 90% compression.

| Model | Compression Strategy | TextVQA | OCRBench | MME | HallB | POPE |
|---|---|---|---|---|---|---|
| Mini-Monkey | Pooling (0.5) | 47.6 | 256 | 1765.2 | 31.5 | 84.5 |
| Mini-Monkey | Random (0.5) | 63.5 | 503 | 1805.5 | 36.2 | 85.9 |
| Mini-Monkey | FastV (Chen et al., 2024a) (0.5) | 73.4 | 781 | 1848.0 | 38.3 | 83.9 |
| Mini-Monkey | FastV (Chen et al., 2024a) (0.9) | 73.9 | 792 | 1866.1 | 37.5 | 85.8 |
| Mini-Monkey | SCM (0.5) | 74.7 | 794 | **1886.0** | **38.7** | 86.1 |
| Mini-Monkey | SCM (0.9) | **75.2** | **801** | 1884.7 | 38.6 | **86.2** |

three components leads to a decline in performance (r2, r4, r5, and r6). The removal of the global component results in the most significant performance drop (r2 and r6). This is because InternVL2-2B was pretrained with both the detailed and global components. Removing the global component or detailed component will result in a significant performance drop. The adaptive component can to some extent compensate for the information provided by the detailed group, thus the impact of removing the detailed component is less significant than the global component. However, to achieve optimal performance, the synergy among all three components is indispensable. These results demonstrate the effectiveness of our method.

**Scale Compression Mechanism (SCM).** We compared SCM with the related work FastV (Chen et al., 2024a). For different methods, we compress the number of visual tokens by 50%. For our method and FastV, we further conduct an experiment with 90% compression. Following FastV's paper, we set the K in FastV as 2. As illustrated in Tab. 9, when using 50% compression and 90% compression, our method outperformed FastV by 21.5% and 4.4%, respectively, demonstrating its effectiveness. FastV compresses input tokens, including both visual and textual tokens, within Transformer blocks. In contrast, our method works in conjunction with the CIP and more target by using tokens with high relative information density to compress tokens with low information density.

**The different compression rates of SCM.** We conduct an ablation experiment on the MME to show how the compression rates of SCM influence the acceleration and computational cost. Following FastV (Chen et al., 2024a), we consider the computation of multi-head attention (MHA) and feed-forward network (FFN) modules in the FLOPs estimation. The total FLOPs are estimated by $L * (4 * n * d^2 + 2 * n^2 * d + 2 * n * d * m)$ where n is the token number, d is the hidden state size, m is the intermediate size of FFN, L is the number of transformer layer. The latency experiments are conducted on A6000 GPU. As presented in Tab. 6, we can find that as the compression ratio increases, the computational load continues to decrease, and the speed keeps improving, without a significant drop in performance. More ablation studies about the selection of hyperparameters are presented in appendix A.

Table 10: Ablation study on the number of sub-images. The number denotes the sub-image count.

| Model | Number | TextVQA | OCRBench | MME | HallB | POPE |
|---|---|---|---|---|---|---|
| Baseline | 18 | 74.2 | 782 | 1851.7 | 37.0 | 85.8 |
| Baseline | 24 | 74.4 | 783 | 1857.6 | 36.9 | 85.8 |
| Baseline | 32 | 74.3 | 782 | 1845.0 | 36.9 | 85.9 |
| Baseline | 48 | 74.0 | 767 | 1841.6 | 36.2 | 85.7 |
| CIP | 32 | **76.0** | **806** | **1884.2** | **38.8** | **88.0** |

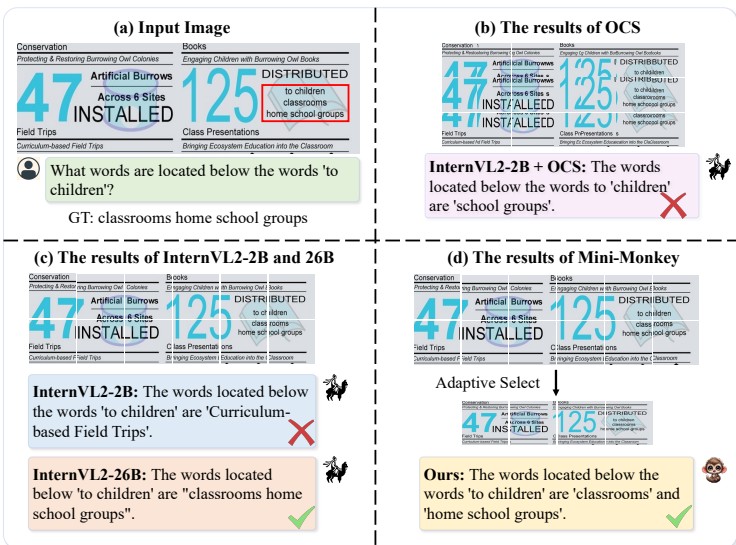

Figure 3: Qualitative results of Mini-Monkey. (a) Input Image and Ground Truth. (b) The results of using overlapping cropping strategy. OSC represents the overlapping cropping strategy. (c) The results of InternVL2-2B and InternVL2-26B. (d) The results of Mini-Monkey.

## 4.4 QUALITATIVE RESULTS

In this section, we provide some qualitative results to demonstrate the effectiveness of our method. First, we verify that the semantic sawtooth effect is particularly evident in lightweight MLLMs, which adopt InternVL2-2B and InternVL2-26B. As shown in Fig. 3(c), InternVL2-26B can answer the questions correctly. However, due to the word 'classrooms' and 'school' being cropped, InternVL2-2B gives a wrong answer that addresses the text in the bottom left corner of the original image. Mini-Monkey can overcome this semantic sawtooth effect and provide the correct answer, as presented in Fig. 3(d). Comparing Fig. 3(b) and Fig. 3(d), we can see that the overlapping cropping strategy introduces some hallucinations and cannot answer questions accurately based on the image, whereas our methods can effectively address the semantic sawtooth effect. More qualitative results are presented in appendix A.7.

## 5 CONCLUSION

In this study, we introduce a Complementary Image Pyramid (CIP) designed to alleviate the semantic sawtooth effect for MLLMs, thereby enhancing their capability to process high-resolution images effectively. CIP is plug-and-play and can be seamlessly integrated into various multimodal large language models at a low cost. We demonstrate the effectiveness of the proposed CIP across diverse architectures, various parameters, and different usage configurations, leading to consistent performance improvements. Besides, we present a Scale Compression Mechanism (SCM) to compress the visual tokens for computational efficiency. CIP not only enhances the general multimodal understanding performance but also shows consistent improvements in document understanding tasks. Furthermore, our experimental results demonstrate that 2B-parameter MLLM equipped with CIP even surpasses larger 8B-parameter state-of-the-art models like InternVL2-8B on the OCRBench.

ACKNOWLEDGMENTS

This research is supported in part by National Natural Science Foundation of China (Grant No.: 62476093, 62441064, 62206104, 62225603, 62441604).

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

# A    APPENDIX

## A.1    ABLATION STUDY OF THE MAXIMUM NUMBER OF TILES.

We conduct an experiment to explore the effect of varying the number of tiles in CIP. The results are presented in Table 11. Our findings indicate that overall performance initially improves with an increase in the number of tiles but begins to decline after reaching a certain point. Optimal performance is achieved when the maximum number of tiles is set to 24. Therefore, we choose 24 as the default maximum number of tiles. Additionally, we perform K-means clustering on the resolution ratios of images and use the clustering results as the predefined aspect ratios. We found that using the clustering results as predefined aspect ratios provides a slight improvement over manually preset them.

Table 11: Ablation study of the maximum number of tiles.

| maximum number of tiles | TextVQA | OCRBench | MME | HallB | POPE |
|---|---|---|---|---|---|
| 48 | 75.4 | 782 | 1837.2 | 39.0 | 87.5 |
| 36 | 75.7 | 784 | 1814.5 | **39.1** | 87.3 |
| 24 | 76.0 | **806** | 1884.2 | 38.8 | 88.0 |
| 12 | 75.5 | 796 | 1874.1 | 38.8 | 87.4 |
| 6 | 74.1 | 788 | 1879.2 | 37.9 | 87.2 |
| K-means | **76.2** | 806 | **1891.5** | 39.1 | **88.1** |

## A.2    ABLATION STUDY OF DIFFERENT SETTINGS OF GROUPING OPERATION.

We conduct experiments to investigate different pre-defined aspect ratio settings for the CIP. All experiments are performed using 24 as the maximum number of tiles $N_{tile}$. The pre-defined aspect ratios are determined according to the following formula:

$$\{g = (n_h \times n_w) | N_{min} \leq n_h \cdot n_w \leq N_{max}, n_h \in \mathbb{N}, n_w \in \mathbb{N}\}.$$

where $n_h$ and $n_w$ represent the height and width of the grid $g$, respectively. the results are shown in the Tab. 12. $\frac{1}{2} < i < 1$ represents the $N_{min}$ is set to $\frac{1}{2} * N_{tile}$ and the $N_{max}$ is set to $1 * N_{tile}$. According to the results of the experiment, we chose $\frac{1}{3} < i < 1$ for detailed group and $\frac{1}{8} < i < \frac{1}{3}$ for adaptive group. In contrast, the global group employs a fixed 1:1 aspect ratio.

Table 12: Ablation study of the different settings of grouping operation.

| Detailed Group | Adaptive Group | TextVQA | OCRBench | MME | HallB | POPE |
|---|---|---|---|---|---|---|
| $\frac{1}{2} < i < 1$ | $\frac{1}{4} < i < \frac{1}{2}$ | 76.0 | 800 | 1886.7 | 38.7 | 87.7 |
| $\frac{1}{3} < i < 1$ | $\frac{1}{4} < i < \frac{1}{3}$ | **76.1** | 804 | 1882.0 | 38.1 | 87.8 |
| $\frac{1}{3} < i < 1$ | $\frac{1}{8} < i < \frac{1}{3}$ | 76.0 | **806** | **1884.2** | **38.8** | **88.0** |
| $\frac{1}{4} < i < 1$ | $\frac{1}{8} < i < \frac{1}{4}$ | 75.7 | 801 | 1873.7 | 38.0 | 87.9 |
| $\frac{3}{4} < i < 1$ | $\frac{1}{8} < i < \frac{3}{4}$ | 75.6 | 798 | 1860.6 | 38.6 | 87.3 |

Table 13: Ablation study on the impact of different numbers of LLM layers in SCM.

| The number of LLM layers | TextVQA | OCRBench | MME | HallB | POPE | Flops (B) | Latency/Example |
|---|---|---|---|---|---|---|---|
| 6 | **75.3** | **798** | **1890.8** | 37.6 | **88.1** | **489.7** | 1.1s |
| 4 | 75.0 | 795 | 1881.2 | 38.6 | 86.1 | 457.0 | 0.99s |
| 2 | 74.7 | 794 | 1886.0 | **38.7** | 86.1 | 424.4 | 0.92s |
| 1 | 74.5 | 789 | 1878.2 | 38.3 | 86.0 | 408.6 | **0.89s** |

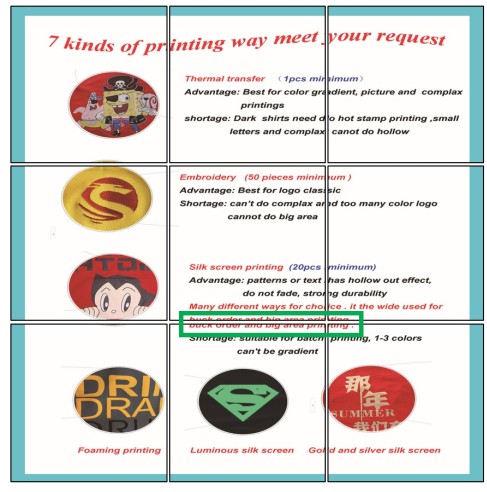 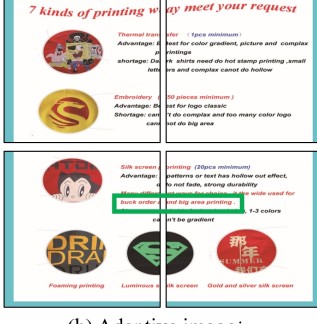 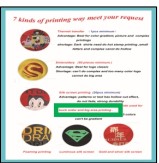

(a) Detail image：
texts have been severely corrupted and are hard to read.

(b) Adaptive image：
texts can be recognized easily

(c) Global image：
texts are blurry

Figure 4: Qualitative results of CIP. The green box indicates the text that needs to be recognized.

## A.3 ABLATION STUDY OF THE DIFFERENT NUMBERS OF LLM LAYERS IN SCM.

To further investigate the effect of varying the number of layers in LLMs on the compression of visual tokens, we conducted a series of experiments. All experiments are conducted in using 0.5 compression rate. The results are detailed in Tab. 13. Our findings indicate that increasing the number of layers leads to enhanced model performance. Nevertheless, this improvement comes at the cost of increased computational demands and higher latency. Balancing these factors, we decided to adopt a two-layer LLM as our standard configuration, optimizing for both efficiency and performance.

## A.4 EFFECTIVENESS OF ADAPTIVE GROUP

To more intuitively demonstrate the role of the adaptive group, we present a visualization case, as shown in Fig. 4. We can find that the texts are severely corrupted and are hard to read in the detailed image. The global image is used to help retain some of the overall context. However, due to the low-resolution of global images, the texts are blurry. The adaptive group is capable of dynamically adjusting based on the needs of the detailed group and provides more fine-grained feature representations. MLLMs can easily read the texts from the adaptive image.

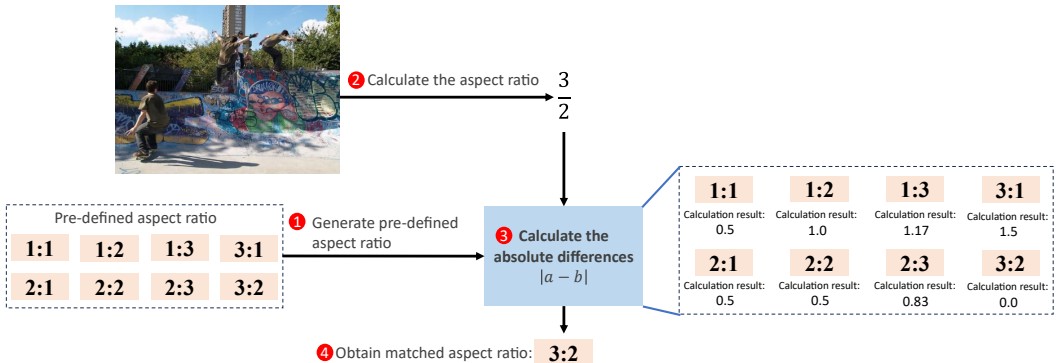

Figure 5: The process of selecting the optimal ratio. First, we generate a set of pre-defined aspect ratios. Next, for a given input image, we calculate its aspect ratio. Then, we calculate the absolute differences between the image's aspect ratio and each of the pre-defined ratios. Lastly, we select the pre-defined aspect ratio that has the smallest difference as the optimal match.

## A.5 THE PROCESS OF SELECTING THE OPTIMAL RATIO

Given an input image, we first compute the absolute differences between its aspect ratio and those within the detailed group: $|a - b|$. The aspect ratio with the smallest absolute difference to that of the input image is then selected as the matched ratio. The overall process is shown in Fig. 5.

## A.6 EXPERIMENT ON VIDEO TASKS.

We also conduct experiments on video tasks, using MMBench-Video (Fang et al., 2024) and Video-MME (Fu et al., 2024). The results are presented in Tab. 14 and Tab. 15. We found that CIP offers only marginal improvements on these two datasets. The primary reason for this is that these video datasets require relatively low-resolution inputs; they typically require small images without the need for resolution enhancement techniques, such as cropping strategies. Therefore, the enhancement brought by CIP on these datasets is limited.

Table 14: Comparison of different methods on MMBench-Video.

| Method | Overall Mean | CP | FP-S | FP-C | HL | Mean | LR | AR | RR | CSR | TR | Mean |
|--------|--------------|-----|------|------|-----|------|-----|-----|-----|-----|-----|------|
| LLaMA-VID (Li et al., 2024d) | 1.08 | 1.30 | 1.09 | 0.93 | 0.42 | 1.09 | 0.71 | 1.21 | 1.08 | 0.83 | 1.04 | 1.02 |
| VideoStreaming (Qian et al., 2024) | 1.12 | 1.38 | 1.13 | 0.8 | 0.32 | 1.13 | 0.77 | 1.27 | 1.11 | 1.01 | 1.10 | 1.09 |
| LLaVA-NeXT-Video (Li et al., 2024a) | 1.14 | 1.35 | 1.15 | 0.97 | 0.58 | 1.14 | 0.64 | 1.38 | 1.30 | 1.27 | 1.03 | 1.13 |
| InternVL2-2B (Baseline) | 1.19 | 1.47 | 1.20 | 1.0 | 0.79 | 1.21 | 0.91 | 1.20 | 1.33 | 1.17 | 1.05 | 1.15 |
| Mini-Monkey-2B (Ours) | 1.20 | 1.45 | 1.22 | 1.06 | 0.74 | 1.22 | 0.89 | 1.19 | 1.42 | 1.17 | 1.05 | 1.16 |

Table 15: Performance of MLLMs on Video-MME with short, medium, and long durations, under the setting of "without subtitles".

| Models | LLM Params | Without Subtitles (%) | | | Overall (%) |
|--------|-----------|------|--------|------|-------------|
| | | Short | Medium | Long | |
| **Open-source Video MLLMs** | | | | | |
| Video-LLaVA (Lin et al., 2023a) | 7B | 45.3 | 38.0 | 36.2 | 39.9 |
| ST-LLM (Liu et al., 2024e) | 7B | 45.7 | 36.8 | 31.3 | 37.9 |
| VideoChat2-Mistral (Li et al., 2024c) | 7B | 48.3 | 37.0 | 33.2 | 39.5 |
| Chat-UniVi-V1.5 (Jin et al., 2024) | 7B | 45.7 | 40.3 | 35.8 | 40.6 |
| **Open & Closed-source Image MLLMs** | | | | | |
| Qwen-VL-Chat (Wang et al., 2024) | 7B | 46.9 | 38.7 | 37.8 | 41.1 |
| Qwen-VL-Max (Wang et al., 2024) | - | 55.8 | 49.2 | 48.9 | 51.3 |
| InternVL-Chat-V1.5 (Chen et al., 2024b) | 20B | 60.2 | 46.4 | 45.6 | 50.7 |
| InternVL2 (Chen et al., 2024b) | 2B | 55.4 | 40.6 | 35.4 | 43.8 |
| Mini-Monkey | 2B | 54.6 | 40.1 | 35.1 | 43.3 |

Table 16: Quantitative accuracy (%) comparison of our model with existing multimodal large language models (MLLMs) on several benchmarks. Following TextMonkey Liu et al. (2024f), we use the accuracy metrics to evaluate our method.

| Method | CIP | Scene Text-Centric VQA | | Document-Oriented VQA | | | KIE | | | OCRBench |
|---|---|---|---|---|---|---|---|---|---|---|
| | | STVQA | TextVQA | DocVQA | InfoVQA | ChartQA | FUNSD | SROIE | POIE | |
| MiniCPM-V-2.6-8B | | 63.8 | **73.4** | **82.0** | 53.7 | 69.2 | 42.9 | 62.1 | 80.4 | 852 |
| MiniCPM-V-2.6-8B | ✓ | **65.3** | 73.1 | 81.7 | **55.3** | **70.6** | **43.4** | **62.5** | **81.1** | **858** |

## A.7 MORE QUANTITATIVE RESULTS

We present several visualization results from Mini-Monkey, as illustrated in Figure 6. In Figure 6(a), we evaluate the model's performance on general multimodal comprehension. When asked about the characters depicted in the image, Mini-Monkey demonstrated its capability by accurately identifying multiple characters from the Avengers.

In Figure 6(b), we tested the model's understanding of contextual scenarios. By posing questions related to the swimming pool setting, Mini-Monkey not only correctly identified the activities taking place but also provided an insightful analysis of potential hazards associated with the environment, showcasing its ability to infer beyond the visible elements.

Figure 6(c) highlights Mini-Monkey's proficiency in extracting structured information from images. We tasked the model with converting the visual data into a JSON format, and it successfully produced a detailed and accurate representation, indicating its strong capacity for data organization and structure.

Finally, in Figure 6(d), we assessed the model's ability to process and analyze menu-related information. Mini-Monkey was not only able to precisely recognize and read the text within the image but also effectively understood the context of the questions posed and performed the required mathematical calculations, thereby demonstrating its comprehensive skill set in combining visual and textual analysis.

## A.8 THE ACCURACY METRIC.

In this section, we detail the metric described in (Liu et al., 2023c), which establishes a uniform and comprehensible standard for assessing text-focused benchmarks. We utilize benchmarks such as ChartQA (Masry et al., 2022), DocVQA (Mathew et al., 2021), InfoVQA (Mathew et al., 2022), TextVQA (Singh et al., 2019), STVQA (Biten et al., 2019), FUNSD (Jaume et al., 2019), SROIE (Huang et al., 2019), and POIE (Kuang et al., 2023). This metric evaluates outcomes by confirming the presence of the ground truth (GT) within the model's output while excluding any responses that are less than four characters in length. To further validate the effectiveness of our proposed CIP, we have also conducted experiments using this metric with the state-of-the-art method, MiniCPM-V-2.6-8B (Yao et al., 2024). As presented in Tab. 16, our findings reveal that even when applied to a leading multimodal large language model like MiniCPM-V-2.6-8B, the proposed CIP can enhance performance. These results further indicate the effectiveness of the proposed CIP.

## A.9 LIMITATIONS.

To ensure the seamless application of our method across various architectures, we adopt an image-centric approach to construct an image pyramid without introducing additional parameters. In future work, we will explore the use of trainable Feature Pyramid Network (FPN) for MLLMs, aiming to more efficiently leverage multi-scale features.

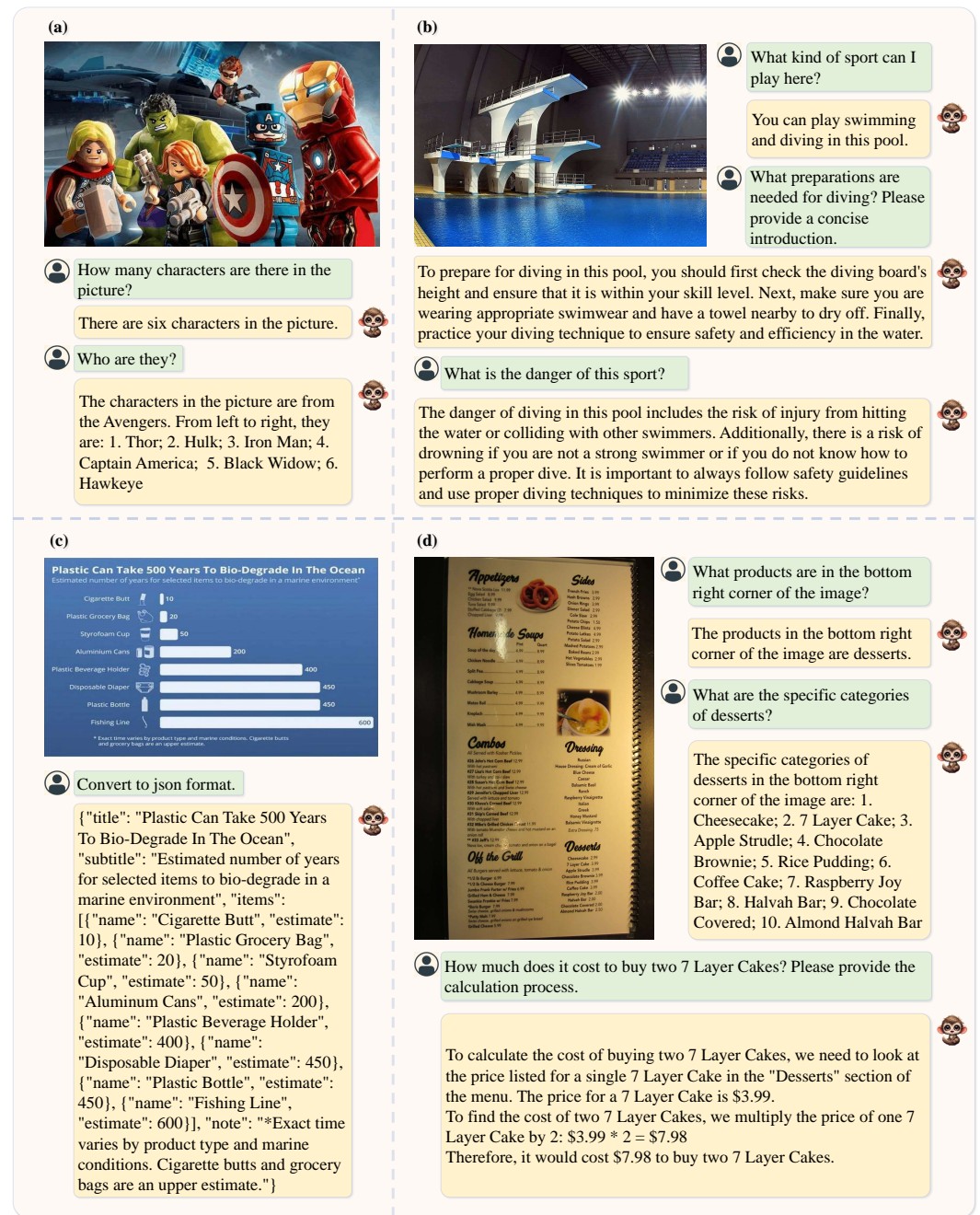

Figure 6: Qualitative results of Mini-Monkey. Figures (a) and (b) pertain to general multimodal understanding. Figures (c) and (d) relate to document understanding.

