# OpenReview forum: "Mini-Monkey: Alleviating the Semantic Sawtooth Effect for Lightweight MLLMs via Complementary Image Pyramid"
_ICLR.cc/2025/Conference — ICLR 2025 Poster_

### Official Review · Reviewer_q5HV · 2024-10-17

**Soundness:** 3
**Presentation:** 2
**Contribution:** 2
**Rating:** 6
**Confidence:** 4

**Summary:**

The paper focuses on the issue of semantic discontinuity in MLLM when scaling images to high resolution, particularly through a sliding-window cropping strategy that can misidentify small or irregularly shaped objects. To tackle this problem, the paper proposes the Complementary Image Pyramid (CIP), which dynamically constructs an image pyramid to enhance semantic information. Besides, the authors introduce a Scale Compression Mechanism (SCM) to minimize computational overhead by compressing redundant visual tokens. Experimental results show the proposed method achieves the best performance across diverse benchmarks.

**Strengths:**

1. Mini-Monkey tackles an important problem in MLLM: scaling images to high resolution. The Complementary Image Pyramid (CIP) introduces the pyramid structure, which is an interesting idea. Besides, CIP and SCM are plug-and-play, which can be easily integrated into different MLLMs.
2. The experiments are sufficient.

**Weaknesses:**

1. Implications of this research. Whether the semantic sawtooth effect mentioned in the paper is a necessary issue to investigate, common Crop-based methods (such as LLaVA-UHD [1] and InternVL [2]) put all cropped regions into a sequence, which does not affect semantic continuity.
2. The work is incremental. The core crop strategy has been widely used in other approaches. CIP is an incremental improvement and doesn't mean much to the community.
3. The architecture is highly sophisticated. The global group is enough to solve the loss of fine-grained features caused by the detailed group. This brings the question of whether the proposed adaptive group is necessary for CIP.
4. The writing needs further improvement. Authors are suggested to improve the readability of the paper. For example, it is hard to understand "For the detailed group, we calculate the aspect ratio of the input image and then compare it with the aspect ratios within the detailed group by calculating the absolute differences." in L231-L233. How to compare? Another example: L270 "We reuse the layer of the LLM as this LLM's Layer". What's the difference between two LLMs?
5. Incomplete experimental analysis. Experimental analysis should include analysis of reasons and not just a list of indicators.

[1] Xu R, Yao Y, Guo Z, et al. Llava-uhd: an lmm perceiving any aspect ratio and high-resolution images[J]. arXiv preprint arXiv:2403.11703, 2024.

[2] Chen Z, Wu J, Wang W, et al. Internvl: Scaling up vision foundation models and aligning for generic visual-linguistic tasks[C]//Proceedings of the IEEE/CVF Conference on Computer Vision and Pattern Recognition. 2024: 24185-24198.

**Questions:**

Please see the weaknesses.

---

> ### Author Response · Authors · 2024-11-21
> **Response to Reviewer q5HV [1/2]**
>
> Thank you for your feedback on our work!
>
> ---
> ***W1. Implications of this research. Whether the semantic sawtooth effect mentioned in the paper is a necessary issue to investigate, common Crop-based methods (such as LLaVA-UHD [1] and InternVL [2]) put all cropped regions into a sequence, which does not affect semantic continuity.***
>
> Thanks for the comments. Simply putting all cropped regions into a sequence is inadequate to alleviate this issue for two primary reasons:
>
> (1) It is worth noting that the visual features in crop-based methods are primarily extracted by ViT. Since each sub-image is encoded independently, there is a lack of feature interaction across different sub-images.
>
> (2) Due to the causal mask in LLM, the earlier features cannot access the later features, resulting in the feature interaction within LLMs is insufficient.
>
> Regarding the two methods mentioned, MiniCPM-V [3] uses the same method as LLaVA-UHD [1], and we have verified CIP's effectiveness on this architecture, as shown in Table 8 in the revised manuscript. For InternVL 2, we have conducted experiments on it as shown in Table 8 in the revised manuscript. Our method provides consistent improvements across these architectures, particularly in OCR-related tasks.
>
> [1] Xu R, Yao Y, Guo Z, et al. Llava-uhd: an lmm perceiving any aspect ratio and high-resolution images[J]. arXiv preprint arXiv:2403.11703, 2024.
>
> [2] Chen Z, Wu J, Wang W, et al. Internvl: Scaling up vision foundation models and aligning for generic visual-linguistic tasks[C]//Proceedings of the IEEE/CVF Conference on Computer Vision and Pattern Recognition. 2024: 24185-24198.
>
> [3] Yao Y, Yu T, Zhang A, et al. Minicpm-v: A gpt-4v level mllm on your phone[J]. arXiv preprint arXiv:2408.01800, 2024.
>
> ---
> ***W3. The architecture is highly sophisticated. The global group is enough to solve the loss of fine-grained features caused by the detailed group. This brings the question of whether the proposed adaptive group is necessary for CIP.***
>
> Thanks for the comments. We would like to clarify that our method is simple yet effective, without incorporating overly sophisticated designs. Specifically:
>
> 1. The CIP is a plug-and-play method that can be adopted as a direct replacement for existing cropping techniques. It can be integrated ***without the need for additional training or modifications*** to the model's architecture, as shown in Table 7 in the revised manuscript.
>
> 2. The SCM utilizes attention weights to compress redundant visual tokens ***without modifying the architecture of the model***.
>
> 3. Both CIP and SCM are ***parameter-free*** and can be ***easily integrated into various MLLMs without introducing additional parameters***, maintaining the overall simplicity of the model.
>
> Overall, the proposed method is both simple and effective, avoiding complex designs. This simplicity is also highlighted by ***Reviewer HRBu, who notes that "... this approach is simpler and more effective, as it does not require additional parameters or training.***"
>
>
> The low-resolution global group helps retain some of the overall context. However, the low-resolution global image is insufficient to provide finer details, due to the low-resolution of the image. In Section A.4 of the appendix in the revised manuscript, we provide more intuitive qualitative results to illustrate the effectiveness of the adaptive group. This limitation is also observed in other works, such as Hugging Face's Idefics3, as discussed in Section 2.2.3 of their paper [1]. In contrast, the adaptive group is capable of dynamically adjusting based on the needs of the detailed group and provides more fine-grained feature representations. Furthermore, as shown in the Table below, removing the adaptive group results in a performance drop. These results demonstrate the effectiveness of the adaptive group.
>
>
> | Method   | TextVQA  | OCRBench | MME     | HallB   | POPE    |
> |----------------------|:----------:|:----------:|:----------:|:----------:|:----------:|
> |  CIP  | 76.0 | 806 | 1884.2 | 38.8 | 88.0 |
> |  Remove Adaptive Group  | 74.6 | 785 | 1853.5  | 37.6  |    87.6     |
>
> [1] Laurençon H, Marafioti A, Sanh V, et al. Building and better understanding vision-language models: insights and future directions[J]. arXiv preprint arXiv:2408.12637, 2024.

---

> ### Author Response · Authors · 2024-11-21
> **Response to Reviewer q5HV [2/2]**
>
> ---
> ***W2. The work is incremental. The core crop strategy has been widely used in other approaches. CIP is an incremental improvement and doesn't mean much to the community.***
>
> We appreciate the Reviewer q5HV for the comment but (strongly) disagree with our highest respect. The proposed CIP is different from the existing cropping strategy. We would like to clarify the novelty of CIP lies in seveal aspect:
>
> 1. The CIP is designed as a plug-and-play module that ***dynamically constructs an complementary image pyramid***. This pyramid provides complementary semantic information to cropping-based MLLMs, enhancing their performance on both general multimodal understanding and document understanding benchmarks. As shown in Table 8 of the revised manuscript, ClP can be ***easily integrated into different MLLMs and consistently enhances performance.***
>
> 2. The proposed CIP can be adopted as a direct replacement for existing cropping techniques to ***improve the performance without requiring training***, as shown in Table 7 in the revised manuscript.
>
> Overall, CIP presents a simple yet effective solution to improving the perceptual capabilities of MLLMs without requiring training. As ***Reviewer HRBu said "The use of the complementary image pyramid (CIP) to replace high-resolution input images with sub-images of various scales is an excellent idea ... this approach is simpler and more effective, without requiring additional parameters or training."*** and ***Reviewer q5HV said "The Complementary Image Pyramid (CIP) introduces the pyramid structure, which is an interesting idea."***, we believe our method is not incremental, and would be interesting and valuable for the community.
>
> ---
> ***W4. The writing needs further improvement. Authors are suggested to improve the readability of the paper. For example, it is hard to understand "For the detailed group, we calculate the aspect ratio of the input image and then compare it with the aspect ratios within the detailed group by calculating the absolute differences." in L231-L233. How to compare? Another example: L270 "We reuse the layer of the LLM as this LLM's Layer". What's the difference between two LLMs?***
>
> Thanks for the comments. We have carefully reviewed and revised the manuscript to poolish the paper.
>
> 1. For the example in L231-L233, we calculate the absolute differences between the aspect ratio of the input image and aspect ratios within the detailed group: $\lvert a - b \rvert $. Then, the ratio that has the smallest absolute difference from the input image's aspect ratio is selected as the matched ratio. A clear illustration of this process is provided in the Section A.5 of the appendix in the revised manuscript.
>
> 2. For the example in L270, the first "LLM" refers to the LLM component of MLLM. The second "this LLM's Layer," refers to the LLM's layer used in the SCM.
>
>
>
> ---
> ***W5. Incomplete experimental analysis. Experimental analysis should include analysis of reasons and not just a list of indicators.***
>
> Thanks for the suggestion. We have included more experimental analysis in Sections 4.2 and 4.3 of the revised manuscript. For instance:
>
> 1. In Section 4.2: "The results indicate that CIP enhances Mini-Monkey's perception ability, thereby improving its capability to handle general multimodal understanding tasks. Additionally, on the POPE benchmark, which evaluates hallucinations in MLLMs, Mini-Monkey outperforms the Baseline InternVL2-2B by 2.8%, demonstrating that CIP can also mitigate hallucinations in MLLMs."
>
> 2. In Section 4.2: "The CIP provides the model with complementary semantic and multi-scale information, enhancing its ability to perceive fine-grained and varying-sized text. With these complementary semantic and multi-scale information, on the OCRBench, Mini-Monkey even surpasses the 8B-parameter Large Multimodal Model InternVL2-8B and the 9B-parameter Large Multimodal Model GLM4-V by 12 and 20, respectively."
>
> 3. In Section 4.2: "OCR-related tasks are utilized to evaluate the fine-grained recognition capabilities of the MLLM. The results from these tasks demonstrate the effectiveness of CIP in enhancing such capabilities."
>
> 4. In Section 4.3: "The results shown in Tab.8 show that ClP can be seamlessly integrated into various MLLMs and consistently improves their performance."
>
>
> ***If our response has answered your question, please consider giving us a higher rating. If you have more questions or need further clarification, please contact us to continue our discussion.***

---

> > ### Comment · Reviewer_q5HV · 2024-11-25
> >
> > Thanks for your reply! I believe the discussions on the concerns above are meaningful, addressing all my questions. I have an additional point and want to discuss with the authors: How about the effect your method applied to video tasks?
> >
> > The above topic is just a discussion and it will be very great if there remains time to conduct the experiments and report the corresponding results. The experiment is not the necessary section. I will change my rating from **borderline reject** to **borderline accept**.

---

> ### Author Response · Authors · 2024-11-25
> **Video Tasks**
>
> We sincerely thank the reviewer for the thoughtful feedback on our work! Regarding video tasks, due to time constraints, we conduct experiments on MMBench-Video [1]. The results are presented in the table below. We find that our method also demonstrates improvements when applied to videos. We will conduct more experiments on video tasks and update them later in our paper. We would greatly appreciate it if you could improve the final rating. Many thanks again!
>
> | Method   | Overall Mean| CP | FP-S | FP-C | HL | Mean | LR | AR | RR | CSR | TR | Mean |
> |----------------------|:----------:|:----------:|:----------:|:----------:|:----------:|:----------:|:----------:|:----------:|:----------:|:----------:|:----------:|:----------:|
> |    LLaMA-VID [2] |   1.08 |1.30 |1.09 |0.93 |0.42 |1.09 |0.71 |1.21| 1.08 |0.83 |1.04 |1.02 |
> | VideoStreaming [3] | 1.12 |1.38 |1.13 |0.8 |0.32 |1.13 |0.77 |1.27 |1.11 |1.01 |1.10 |1.09 |
> |   LLaVA-NeXT-Video [4] |  1.14 | 1.35 |1.15 |0.97 |0.58 |1.14 |0.64 |1.38 |1.30 |1.27 |1.03 |1.13 |
> |  InternVL2-2B (Baseline)  | 1.19 | 1.47 | 1.20 | 1.0 | 0.79 | 1.21 | 0.91 | 1.20 | 1.33 |1.17 | 1.05 | 1.15 |
> |  Mini-Monkey-2B (Ours)  | 1.20 | 1.45 | 1.22 | 1.06 | 0.74 | 1.22 | 0.89 | 1.19 | 1.42 |1.17 | 1.05 | 1.16 |
>
> [1] Fang X, Mao K, Duan H, et al. MMBench-Video: A Long-Form Multi-Shot Benchmark for Holistic Video Understanding[J]. arXiv preprint arXiv:2406.14515, 2024.
>
> [2] Li Y, Wang C, Jia J. Llama-vid: An image is worth 2 tokens in large language models[C]//European Conference on Computer Vision. Springer, Cham, 2025: 323-340.
>
> [3] Qian R, Dong X, Zhang P, et al. Streaming long video understanding with large language models[J]. arXiv preprint arXiv:2405.16009, 2024.
>
> [4] Zhang Y, Li B, Liu H, et al. Llava-next: A strong zero-shot video understanding model[J]. 2024.

---

> > ### Comment · Reviewer_q5HV · 2024-11-25
> >
> > Thanks for your quick reply. Good job!

---

### Official Review · Reviewer_HRBu · 2024-10-26

**Soundness:** 3
**Presentation:** 3
**Contribution:** 3
**Rating:** 6
**Confidence:** 4

**Summary:**

Existing multimodal large language models (MLLMs) often use cropping when processing high-resolution images (divides the high-res image into multiple lower-resolution as the input). However, Non-Overlapping Cropping can lead to semantic discontinuity and semantic damage, referred to by the authors as the "semantic sawtooth effect." On the other hand, Overlapping Cropping results in redundant visual information.

To address this, the paper proposes a complementary image pyramid, which aims to alleviate the semantic sawtooth effect in the context of Non-Overlapping Cropping. To mitigate the additional computational burden introduced by this module, the authors propose a Scale Compression Mechanism. This mechanism leverages the attention weights of the LLM and the proposed multi-scale image semantics in a training-free and parameter-free manner to compress redundant tokens.

The proposed approach achieves promising results on 8 general multimodal understanding benchmarks and 9 document understanding benchmarks.

**Strengths:**

1. The use of the complementary image pyramid (CIP) to replace high-resolution input images with sub-images of various scales is an excellent idea. Compared to existing methods that introduce multi-scale visual signals through models, this approach is simpler and more effective, without requiring additional parameters or training.

2. The Scale Compression Mechanism (SCM) reasonably and interestingly reduces the extra computational load brought by multi-scale input sub-images by compressing visual tokens.

3. Both the proposed CIP and SCM do not require the introduction of additional parameters or training, making them applicable to different MLLMs.

4. The method proposed in this paper achieves promising results in various general multimodal understanding and document understanding benchmarks.

**Weaknesses:**

1. The authors claim in lines 265-266 that "a well-trained LLM from MLLM can effectively select the necessary visual features based on the input question," which seems to differ from the conclusions of existing MLLM works [1,2]. This discrepancy makes me question the effectiveness of the proposed method. If MLLMs cannot truly understand images, how can their attention weights be used here to compress visual tokens?

2. The rationale for selecting only the first and second layers of the LLM to choose visual tokens is not sufficiently explained, and no ablation studies have been conducted. How would the results differ if more LLM layers were selected, only one LLM layer was chosen, or the selection was done randomly without using LLM attention priors? In conclusion, ablation studies have only been conducted on the Resolution Strategy, lacking ablation experiments on the compression of visual tokens.

3. For the complementary image pyramid, the authors need to manually preset a set of predefined aspect ratios, which seems somewhat tricky. How these aspect ratios are set and why these specific values are chosen remains unclear. A better solution might be to perform K-means clustering on the resolution ratios of images and use the clustering results as the predefined aspect ratios.

4. When comparing with other methods and conducting ablation studies, only the number of parameters and performance are shown, lacking comparisons on FLOPs. Although your proposed multi-scale input does not introduce new parameters, it does increase actual computational load. Therefore, in the ablation studies of Table 4, the actual computational load and inference overhead should also be compared.

Reference
* [1] Cambrian-1: A Fully Open, Vision-Centric Exploration of Multimodal LLMs, NeurIPS 2024
* [2] Are We on the Right Way for Evaluating Large Vision-Language Models?, NeurIPS 2024

**Questions:**

Please refer to the Weaknesses.

---

> ### Author Response · Authors · 2024-11-20
> **Response to Reviewer HRBu [1/2]**
>
> Thank you very much for your thoughtful feedback on our work!
>
> ---
> ***W1. The authors claim in lines 265-266 that "a well-trained LLM from MLLM can effectively select the necessary visual features based on the input question," which seems to differ from the conclusions of existing MLLM works [1,2]. This discrepancy makes me question the effectiveness of the proposed method. If MLLMs cannot truly understand images, how can their attention weights be used here to compress visual tokens?***
>
> Thanks for the comments! This is worth further discussion. We would like to note that the existing MLLM works [1,2] do not claim that MLLMs cannot understand images. They demonstrate that for a portion of Q&A in some benchmarks, it's possible to answer without the input of the images. However, for most of the Q&A in the majority of datasets, MLLMs need to understand the images to provide correct answers. Furthermore, these studies [1,2] also introduce a new evaluation benchmark, whose results confirm the capability of MLLMs to understand images. Additionally, some works, such as FastV [3], have also demonstrated that an LLM from a well-trained MLLM can effectively select the necessary visual features based on the input question.
>
> [1] Cambrian-1: A Fully Open, Vision-Centric Exploration of Multimodal LLMs, NeurIPS 2024
>
> [2] Are We on the Right Way for Evaluating Large Vision-Language Models?, NeurIPS 2024
>
> [3] Chen L, Zhao H, Liu T, et al. An image is worth 1/2 tokens after layer 2: Plug-and-play inference acceleration for large vision-language models, ECCV 2025.
>
> ---
> ***W2. The rationale for selecting only the first and second layers of the LLM to choose visual tokens is not sufficiently explained, and no ablation studies have been conducted. How would the results differ if more LLM layers were selected, only one LLM layer was chosen, or the selection was done randomly without using LLM attention priors? In conclusion, ablation studies have only been conducted on the Resolution Strategy, lacking ablation experiments on the compression of visual tokens.***
>
> Thanks for the comments! We have investigated the impact of randomly selecting tokens, as presented in the second row of Table 7 in the submitted manuscript. To further investigate the effect of varying the number of layers in LLMs on the compression of visual tokens, we conducted a series of experiments. All experiments are conducted using a 0.5 compression rate. The results are detailed in the Table below. Our findings indicate that increasing the number of layers leads to enhanced model performance. Nevertheless, this improvement comes at the cost of increased computational demands and higher latency. Balancing these factors, we adopt two layers of LLM as our standard configuration, optimizing for both efficiency and performance. We have supplemented these results in Section A.3 of the appendix in the revised manuscript.
>
> | The number of LLM layers  | TextVQA  | OCRBench | MME     | HallB   | POPE    | Flops (B)   | Latency/Example    |
> |----------------------|:----------:|:----------:|:----------:|:----------:|:----------:|:----------:|:----------:|
> |  6  | ***75.3*** | ***798*** | ***1890.8***  | 37.6  |    ***88.1***     | 489.7  |    1.1s     |
> |  4  | 75.0 | 795 | 1881.2  | 38.6  |    86.1     | 457.0  |    0.99s     |
> |  2  | 74.7 | 794 | 1886.0 | ***38.7*** | 86.1 | 424.4  |    0.92s     |
> |  1  | 74.5 | 789 | 1878.2  | 38.3  |    86.0     | 408.6 |    0.89s     |
> | Randomly Selecting Tokens | 63.5 | 503 | 1805.5 | 36.2 | 85.9 | ***392.8*** |    ***0.87s***     |

---

> ### Author Response · Authors · 2024-11-20
> **Response to Reviewer HRBu [2/2]**
>
> ---
> ***W3. For the complementary image pyramid, the authors need to manually preset a set of predefined aspect ratios, which seems somewhat tricky. How these aspect ratios are set and why these specific values are chosen remains unclear. A better solution might be to perform K-means clustering on the resolution ratios of images and use the clustering results as the predefined aspect ratios.***
>
> Thanks for the suggestion! For the setting of predefined aspect ratios, the answer can be found in Q2.2 of Reviewer cDq9. We have revised the description to make it more clear in the revised manuscript. We think this is a good idea to use K-means clustering on the resolution ratios of images and use the clustering results as the pre-defined aspect ratios. Following the suggestion, we conduct an experiment, and the results are shown below. Our findings indicate that using the clustering results as pre-defined aspect ratios yields better performance compared to manually setting them. We have supplemented these results in Table 11 of the appendix in the revised manuscript.
>
> | maximum number of tiles  | TextVQA  | OCRBench | MME     | HallB   | POPE    |
> |----------------------|:----------:|:----------:|:----------:|:----------:|:----------:|
> |  24  | 76.0 | 806 | 1884.2 | 38.8   | 88.0 |
> |  K-means  | ***76.2*** | ***806*** | ***1891.5***  | ***39.1***  |    ***88.1***     |
>
> ---
> ***W4. When comparing with other methods and conducting ablation studies, only the number of parameters and performance are shown, lacking comparisons on FLOPs. Although your proposed multi-scale input does not introduce new parameters, it does increase the actual computational load. Therefore, in the ablation studies of Table 4, the actual computational load and inference overhead should also be compared.***
>
> Thanks for the suggestion! Following the suggestion, we add the computational load and inference overhead in Table 4. The experiments of latency are conducted on a single A6000 GPU. The results are shown in the Table below. Our method outperforms the existing multi-scale strategy by an average of 14 in terms of the corresponding metric with fewer FLOPs and lower latency. We have supplemented the results in Table 4 in the revised manuscript.
>
> | Model  | Resolution Strategy | TextVQA  | OCRBench | MME     | HallB   | POPE    | Flops(B)    | Latency/Example    |
> |----------------------|----------|:----------:|:----------:|:----------:|:----------:|:----------:|:----------:|:----------:|
> |  Baseline  | Dynamic high-res Strategy | 73.4 |784 |1876.8 |37.9 |85.2 | 349.4 | 1.0s
> |  Baseline  | Fixed Size high-res Strategy | 74.2 | 772 | 1824.5 | 37.6 | 85.0 | 510.9 | 1.1s
> |  Baseline  | Overlapping Cropping Strategy | 70.6 | 758 | 1874.1 | 36.8 | 83.5 | 393.1 | 1.1s
> |  Baseline  | Multi-Scale Strategy | 74.8 | 776 | 1846.8 | 38.1 | 85.3 | 559.2 | 1.6s
> |  Ours  | Complementary Image Pyramid | 76.0 | 806 | 1884.2 | 38.8 | 88.0 | 531.3 | 1.3s
>
> ***If our response has answered your question, we would be grateful if you consider giving us a higher rating. If you still have any questions or need further clarification, please reach out to us and continue the discussion.***

---

> ### Author Response · Authors · 2024-11-25
> **Reminder of the Discussion Period Deadline**
>
> Dear Reviewer HRBu,
>
> Thank you for your time and valuable feedback. As the ICLR public discussion phase will be ending on November 26th, we remain open to addressing any remaining questions or concerns. We would greatly appreciate it if you could consider improving the evaluation after reviewing our responses. Thank you very much for your consideration.
>
> Sincerely, Paper 9908 Authors

---

> > ### Comment · Reviewer_HRBu · 2024-11-25
> >
> > Thank you for your response and rebuttal. I am pleased with the answers to most of the questions, and consequently, I have decided to raise the score to 6: marginally above the acceptance threshold.
> >
> > However, I remain confused about the first and most critical issue: if many questions can be answered correctly without viewing the images, it suggests that the MLLM does not need to rely on the images for these questions. In that case, why can the MLLM's attention weights prior be used to compress visual tokens? For these images, it seems that visual tokens might not be necessary at all.

---

> ### Author Response · Authors · 2024-11-26
> **Discussion**
>
> We sincerely appreciate your constructive comments and suggestions. Our work primarily focuses on scenarios where MLLMs need to understand images in order to answer questions. For instance, when presented with an image containing text, if we ask the MLLMs what text is written within it, the MLLMs need to view the image to recognize the textual content. In such cases, the MLLMs' attention weights are effective at compressing visual tokens. For questions that can be answered without input images, the response typically relies on the knowledge learned by the LLM. Whether an image is provided or not, for this type of question, the result may be the same. We agree that when faced with these types of questions, MLLMs' attention weights might fail to handle this case. This presents an interesting challenge regarding token compression in such scenarios. Thank you for your valuable insights and we will explore this area in future research.

---

### Official Review · Reviewer_bHqc · 2024-10-28

**Soundness:** 3
**Presentation:** 3
**Contribution:** 2
**Rating:** 6
**Confidence:** 3

**Summary:**

This paper proposes Mini-Monkey, a lightweight multimodal large language model that effectively mitigates the semantic sawtooth effect in high-resolution image processing through a Complementary Image Pyramid (CIP) and a Scale Compression Mechanism(SCM), achieving superior performance across various benchmarks.

**Strengths:**

1、The paper describes CIP for dynamic segmentation of images and SCM for compression of visual tokens to address the semantic sawtooth effect in MLLM high-resolution image processing, demonstrating innovations in addressing specific challenges.
2、In the CIP module, the model focuses on the feature interactions of different sub-images
3、In the SCM module, the model selectively compresses visual tokens. The interaction information of different types of visual tokens is also considered.

**Weaknesses:**

1、In Figure 2a, the pixel shuffle operation appears, but the paper does not reflect the transformation of the image features before and after this operation.
2、According to the formula in line 241, the aspect ratios of the adaptive and detailed groups are not integer multiples. But for Figure 2b, the final selection of Ah is 1 and Dh is 3, which seems to be a contradiction.
3、In the CIP module, the paper does not present a clear picture of how the predefined slice ratios appropriate to the size of the image are selected, i.e., what principle is it based on.

**Questions:**

1、I would like to inquire why the paper does not mention the maximum resolution of the images that the model supports, as well as the corresponding comparative experiments.

---

> ### Author Response · Authors · 2024-11-20
> **Response to Reviewer bHqc**
>
> Thank you very much for your thoughtful feedback on our work!
>
> ---
> ***W1. In Figure 2a, the pixel shuffle operation appears, but the paper does not reflect the transformation of the image features before and after this operation.***
>
> The pixel shuffle operation is utilized to reduce the number of visual tokens to one-quarter of the original. We have clarified these points in Figure 2a in our revised manuscript.
>
> ---
> ***W2. According to the formula in line 241, the aspect ratios of the adaptive and detailed groups are not integer multiples. But for Figure 2b, the final selection of Ah is 1 and Dh is 3, which seems to be a contradiction.***
>
> Thanks for pointing out this issue! The aspect ratios for the adaptive and detailed groups are set to non-integer multiples to ensure that the cropping lines within each group do not overlap. When the $A_h$ is 1, it means that the cropping operation is not required. Therefore, they can be integer multiples in such cases. We have updated the formula on line 231 to enhance clarity in our revised manuscript.
>
> \begin{equation}
> \forall k \in \mathbb{Z},\, \forall i \in \\{h, w\\}\,
> \begin{cases}
> D_i = k \cdot A_i, & \text{if } A_i = 1, \\\\
> D_i \neq k \cdot A_i, & \text{otherwise.}
> \end{cases}
> \end{equation}
>
> ---
> ***W3. In the CIP module, the paper does not present a clear picture of how the predefined slice ratios appropriate to the size of the image are selected, i.e., what principle is it based on.***
>
> Given an input image, we first calculate its aspect ratio. Then, we calculate the absolute differences between the image’s aspect ratio and each of the pre-defined ratios by $\lvert a - b \rvert $. The aspect ratio with the smallest absolute difference to that of the input image is then selected as the optimal ratio. A clear picture of this process is provided in Section A.5 of the appendix in the revised manuscript.
>
> ---
> ***Q1. I would like to inquire why the paper does not mention the maximum resolution of the images that the model supports, as well as the corresponding comparative experiments.***
>
> Thanks for the comments! As noted on line 305 of the revised manuscript, we limit the maximum number of sub-images to 24. Similar to [1,2], our model could support image resolutions up to 4K. However, when the resolution is increased to a certain point, further increasing the resolution leads to an increase in computational cost without providing a corresponding improvement in performance. We provide an experiment to explore the impact of the maximum resolution of the images in CIP. The results are presented in the Table below. The overall performance shows a trend of first increasing and then decreasing as the maximum resolution increases. Based on these findings, we have chosen 24 as the default configuration. We add these results in Section A.1 of the appendix in the revised manuscript.
>
> | maximum number of tiles  | TextVQA  | OCRBench | MME     | HallB   | POPE    |
> |----------------------|:----------:|:----------:|:----------:|:----------:|:----------:|
> |  48  | 75.6 | 792 | 1837.2  | 39.0  |    87.5     |
> |  36  | 75.7 | 794 | 1814.5  | ***39.1***  |    87.3     |
> |  24  | ***76.0*** | ***806*** | ***1884.2*** | 38.8   | ***88.0*** |
> |  12  | 75.5 | 796 | 1874.1  | 38.8  |    87.4     |
> |  6  | 74.1 | 788 | 1879.2  | 37.9  |    87.2     |
>
> [1] Chen Z, Wang W, Tian H, et al. How far are we to gpt-4v? closing the gap to commercial multimodal models with open-source suites[J]. arXiv preprint arXiv:2404.16821, 2024.
>
> [2] Dong X, Zhang P, Zang Y, et al. Internlm-xcomposer2-4khd: A pioneering large vision-language model handling resolutions from 336 pixels to 4k hd[J]. arXiv preprint arXiv:2404.06512, 2024.
>
> ***Hope that our response has answered your question. If you still have any questions or need more help, we look forward to your response so we can continue the discussion.***

---

### Official Review · Reviewer_cDq9 · 2024-10-29

**Soundness:** 3
**Presentation:** 2
**Contribution:** 3
**Rating:** 6
**Confidence:** 4

**Summary:**

The paper introduces the "semantic sawtooth effect" caused by common cropping strategies in high-resolution image scaling for MLLMs. To tackle this issue, they propose a Complementary Image Pyramid (CIP), a flexible and easy-to-integrate approach aimed at reducing semantic discontinuity by providing rich semantic information across different scales. Alongside CIP, they also introduce a Scale Compression Mechanism (SCM) to minimize computational overhead by compressing unnecessary visual tokens. These enhancements improve performance across various MLLM architectures and capacities, leading to the development of a lightweight model called Mini-Monkey, which shows notable improvements in multimodal and document understanding tasks.

**Strengths:**

1. The proposed CIP is logically clear and reasonable. Experiments show that CIP outperforms other cropping methods.
2. The experiments are comprehensive, showing significant improvements across various model families and sizes, as well as multiple datasets, which demonstrates the effectiveness of the proposed CIP.
3. The paper is well-written and clear.

**Weaknesses:**

1. The insight and inspiration of proposed Adaptive Group is not clear.

2. Lack ablation studies on proposed CIP and SCM

See below for details.

**Questions:**

1. There is a lack of explanation for the setting of the Adaptive Group. While both detail and global are easy to understand, the paper does not explicitly state the benefits of detail group or provide experimental evidence to support it.
2. The experiments lack ablation studies, such as removing the detail, adaptive, or global components. What would be the impact if one of these were removed? Which component is most critical? Secondly, what impact does the number of tiles in CIP have? How does the performance of CIP change with the number of tiles in the detail component? Also, what aspect ratios are set by default for CIP?
3. The motivation behind SCM does not align well with the experiments. The paper mentions that "certain scenarios may restrict the level of computational resources available," but the experimental part does not provide experiments on how different compression rates of SCM affect model acceleration and computational cost.

If the authors could supplement their experiments, I would be willing to raise the score to above borderline.

---

> ### Author Response · Authors · 2024-11-20
> **Response to Reviewer cDq9 [1/3]**
>
> Thank you very much for your thoughtful feedback on our work! Because we utilize the answers from Q2.1 and Q2.2 in responding to Q1, we will first respond to Q2.1 and Q2.2.
>
> ---
> ***Q2.1. The experiments lack ablation studies, such as removing the detail, adaptive, or global components. What would be the impact if one of these were removed? Which component is most critical?***
>
> Thanks for the suggestion! To evaluate the importance of each component in the CIP, we performed ablation studies using the InternVL2-2B. The results are presented in Tabel below. The analysis reveals:
>
> 1. Utilizing either the global component alone or the detailed component alone results in a performance drop, as shown in r1 and r3 of Table. By comparing the r1 and r2, as well as r3 and r4, in Table, we find that adding an Adaptive component improves the performance.
>
> 2. When using both the detailed component and the global component, adding the adaptive component leads to further improvements, as shown in r5 and r6 of Table.
>
> 3. The results indicate that the removal of any one of the three components leads to a decline in performance (r2, r4, r5, and r6). The removal of the global component results in the most performance drop (r2 and r6). This is because InternVL2-2B was pretrained with both the detailed and global components. Removing the global component or detailed component will result in a performance drop. The adaptive component can to some extent compensate for the information provided by the detailed group, thus the impact of removing the detailed component is less than the global component. However, to achieve optimal performance, the synergy among all three components is indispensable.
>
> We added the results of removing different components in CIP in Table 5 of the revised manuscript.
>
> |    |Method   | Global Component   | Detailed Component   | Adaptive Component   | TextVQA  | OCRBench | MME     | HallB   | POPE    |
> |----------------------|----------------------|:----------:|:----------:|:----------:|:----------:|:----------:|:----------:|:----------:|:----------:|
> | r1 |InternVL2-2B | | √ |  | 62.5 | 385 | 1686.2 | 34.8 | 81.8 |
> | r2 |InternVL2-2B | | √ | √ | 70.5 | 473 | 1806.1  | 37.4  | 86.0  |
> | r3 |InternVL2-2B |√ | |  | 60.8 | 624 | 1842.3 | 37.4 | 85.3 |
> | r4 |InternVL2-2B |√ | | √ | 74.8 | 782 | 1874.2  | 39.0  | 87.5  |
> | r5 |InternVL2-2B | √ | √ |  | 74.6 | 785 | 1853.5  | 37.6  |    87.6     |
> | r6 |InternVL2-2B | √ | √ | √ | ***76.0*** | ***806*** | ***1884.2*** | ***38.8*** | ***88.0*** |

---

> ### Author Response · Authors · 2024-11-20
> **Response to Reviewer cDq9 [2/3]**
>
> ---
> ***Q2.2. What impact does the number of tiles in CIP have? How does the performance of CIP change with the number of tiles in the detail component? Also, what aspect ratios are set by default for CIP?***
>
> We conduct an experiment to explore the effect of varying number of tiles in CIP. The results are presented in the Table below. Our findings indicate that overall performance initially improves but then declines as the maximum resolution is increased. Based on this observation, we selected 24 as the default maximum number of tiles. We have added these results in Section A.1 of the appendix in the revised manuscript.
>
>
> | maximum number of tiles  | TextVQA  | OCRBench | MME     | HallB   | POPE    |
> |----------------------|:----------:|:----------:|:----------:|:----------:|:----------:|
> |  48  | 75.4 | 782 | 1837.2  | 39.0  |    87.5     |
> |  36  | 75.7 | 784 | 1814.5  | ***39.1***  |    87.3     |
> |  24  | ***76.0*** | ***806*** | ***1884.2*** | 38.8 | ***88.0*** |
> |  12  | 75.5 | 796 | 1874.1  | 38.8  |    87.4     |
> |  6  | 74.1 | 788 | 1879.2  | 37.9  |    87.2     |
>
> The default aspect ratios used in the CIP are detailed in lines 220 to 229 of the paper. We conduct experiments to investigate different pre-defined aspect ratio settings for the CIP. We set the pre-defined aspect ratios by : $$ \\{ g = (n_h \times n_w) | N_{min} \leq n_h \cdot n_w \leq N_{max}, n_h \in \mathbb{N}, n_w \in \mathbb{N} \\}$$
>
> where $n_h$ and $n_w$ represent the height and width of the grid $g$, respectively.
>
> The results are shown in the Table below. $\frac{1}{2}$ < i < 1 represents the $N_{min}$ is set to $ \frac{1}{2} * N_{tile}$ and the $N_{max}$ is set to $ 1 * N_{tile}$.  All experiments are performed using 24 as the maximum number of tiles $ N_{tile}$. According to the results of the experiment, we chose $\frac{1}{3}$ < i < 1 for detailed group and $\frac{1}{8}$ < i < $\frac{1}{3}$ for adaptive group. In contrast, the global group employs a fixed 1:1 aspect ratio. We have added these results in Section A.2 of the appendix in the revised manuscript.
>
> | Detailed  Group| Adaptive Group | TextVQA  | OCRBench | MME     | HallB   | POPE    |
> |----------------------|----------|:----------:|:----------:|:----------:|:----------:|:----------:|
> |  $\frac{1}{2}$ < i < 1  |  $\frac{1}{4}$ < i < $\frac{1}{2}$ | 76.0 | 800 | ***1886.7*** | 38.7 |  87.7 |
> |  $\frac{1}{3}$ < i < 1 |  $\frac{1}{4}$ < i < $\frac{1}{3}$ | ***76.1*** | 804 | 1882.0 | 38.1 | 87.8 |
> |  $\frac{1}{3}$ < i < 1 |  $\frac{1}{8}$ < i < $\frac{1}{3}$ |76.0 | ***806*** | 1884.2 | ***38.8*** | ***88.0*** |
> |  $\frac{1}{4}$ < i < 1 |  $\frac{1}{8}$ < i < $\frac{1}{4}$ | 75.7 | 801 | 1873.7  | 38.0  |  87.9  |
> |  $\frac{3}{4}$ < i < 1 |  $\frac{1}{8}$ < i < $\frac{3}{4}$ |75.6 | 798 | 1860.6  | 38.6  |  87.3 |

---

> ### Author Response · Authors · 2024-11-20
> **Response to Reviewer cDq9 [3/3]**
>
> ---
> ***Q1. There is a lack of explanation for the setting of the Adaptive Group. While both detail and global are easy to understand, the paper does not explicitly state the benefits of the detail group or provide experimental evidence to support it.***
>
> For the setting of the adaptive group, the pre-defined aspect ratios span from one-eighth to one-third of the maximum number of tiles. We will discuss this point in detail in Q2.2. The adaptive group mainly introduces three benefits:
>
> 1. Cross-Tile Interaction: The adaptive component provides cross-tile interaction features and the cropping positions information for the detailed component. When selecting an aspect ratio, the adaptive component avoids using ratios that are simple multiples of the detailed component's aspect ratio. This ensures that the cropping positions in the detailed component will not be cut in the adaptive component. Therefore, the interactions of the cropping positions and the interactions of different tiles in the detailed component will be supplemented by the adaptive component. In Section A.4 of the appendix in the revised manuscript, we provide more intuitive qualitative results to illustrate the effectiveness of the adaptive group. Similarly, the global component provides the cross-tile interaction features and the cropping positions information for the adaptive component. Three components provide complementary semantic information for the model.
>
> 2. Multi-Scale Information: The adaptive component, together with the detailed component and the global component, offers multi-scale information, enabling the model to better handle objects of different sizes in images.
>
> 3. Plug-and-Play Integration: The adaptive component is plug-and-play, requiring no additional parameters. It can be seamlessly integrated with existing MLLMs that utilize cropping strategies. It can be utilized without training and its effectiveness can be further improved through fine-tuning.
>
> The benefits of detail group: The detail group employs a high resolution to supply the model with fine-grained information, thereby enhancing its ability to perceive small objects or text. Regarding the experimental findings, refer to the third row of the Table in Q2.1. Our observations show that eliminating the detailed component leads to a decline in performance, particularly in text-related tasks that require fine-grained information.
>
> ---
> ***Q3. How different compression rates of SCM affect model acceleration and computational cost.***
>
> Thanks for the suggestion! For the computational cost, following FastV[1], we consider the computation of multi-head attention (MHA) and feed-forward network (FFN) modules in the FLOPs estimation. The total FLOPs are estimated by $L * (4*n*d^2 +2*n^2*d+2*n*d*m) $ where n is the token number, d is the hidden state size, m is the intermediate size of FFN, L is the number of transformer layer. We conducted an experiment on the MME, and the results are presented in the table below. We can find that as the compression ratio increases, the computational load continues to decrease, and the speed keeps improving, without a significant drop in performance. The latency experiments are conducted on a single A6000 GPU. We have added these results in Table 6 of the revised manuscript.
>
> | Compression Rate  | 0.0  | 0.1 | 0.2     | 0.3   | 0.4     | 0.5   | 0.7 | 0.9 |
> |----------------------|:----------:|:----------:|:----------:|:----------:|:----------:|:----------:|:----------:|:----------:|
> | MME | 1884.2 | 1884.7 | 1879.8  | 1878.5  | 1876.3  | 1886.0  |  1871.7 | 1870.2 |
> |  Flops (B)  | 446.9 | 414.9 | 383.6  | 353.0 |  323.0  | 293.7 |    237.0  | 171.4
> |  Latency/Example  | 0.83s | 0.78s | 0.73s  | 0.67s  |    0.63s     |0.59s  |    0.51s   |    0.49s     |
>
>
> [1] Chen L, Zhao H, Liu T, et al. An image is worth 1/2 tokens after layer 2: Plug-and-play inference acceleration for large vision-language models[C]//European Conference on Computer Vision. Springer, Cham, 2025: 19-35.
>
> ***If our response has adequately answered your question, please consider giving us a higher rating. If you still have any doubts or further questions, we are looking forward to continuing the discussion.***

---

> > ### Comment · Reviewer_cDq9 · 2024-11-22
> >
> > Thanks for authors detailed experiments and clarification. As far as I concern, this is a technically-valid and valuable paper, and help contribution to research in image preprocessing strategies and real-world VLM design. In summary, I will raise my score to above borderline.

---

> > > ### Author Response · Authors · 2024-11-22
> > > **Thank you for your response**
> > >
> > > We sincerely thank the reviewer for the constructive feedback and support. Your comments are valuable for us to improve the quality of this work.  We would greatly appreciate it if you could improve the final rating. Many thanks again!

---

### Author Response · Authors · 2024-11-21
**General Response**

Dear reviewers and AC,

We would like to express our gratitude to the reviewers for their diligent efforts in reviewing our work.

As highlighted by the reviewers, we believe our paper proposes an interesting (q5HV, HRBu) and effective (cDq9, HRBu, bHqc) method that alleviates the issue in the exsisting cropping strategies, which can be easily integrated into different MLLMs (cDq9, HRBu, q5HV).

We appreciate your helpful suggestions on our manuscript. In accordance with your comments, we have carefully revised the manuscript with the following additional discussions and experiments:

For experiments:

* We added the results of removing different components in CIP (Table 5);

* We added the results of the various aspect ratio settings in the CIP (Table 11 and Table 12);

* We added the results of ablation studies of SCM (Table 6 and Table 13);

We hope the added experiments will provide more insights. For discussion and analysis:

* We added more details about the CIP to make it clearer (Section 3.1).

* We provide a more intuitive example in Section A.4 of the appendix to demonstrate the effectiveness of the adaptive group (Figure 4);

* We provide a example of the matching process of CIP in Section A.5 of the appendix (Figure 5);

We highlighted the revised contents in blue for your convenience to check. We sincerely believe that Mini-Monkey will be of strong interest to the ICLR community, especially, as the revision allows us to better deliver the effectiveness of our method.

For other concerns, we addressed them in our responses.

Thank you very much!

Authors.

---

### Meta-Review · Area_Chair_DqyP · 2024-12-17

**Metareview:**

This paper tackles the high resolution image requirement in MLLM, and propose a simple, plug-and play method named Complementary Image Pyramid (CIP) to mitigate the semantic sawtooth effect of patch tileing. The reviewers approve for the effectiveness of the proposed method, and detailed ablation study may help the MLLM community to help for image preprocessing. The AC	recommend for accept based on the reviewers’ opinion.

**Additional Comments On Reviewer Discussion:**

This is a borderline paper, most reviewers anticipate the discussion and the authors provide detailed experimental results and discussion in the rebuttal phrase, the overall method is simple but effective, and can be used as plug-in modules for image preprocessing in MLLM. The AC suggest to compare or discuss the dynamic tiling method with recent naïve high resolution strategy such as NaViT used in Qwen-VL.

---

### Decision · Program_Chairs · 2025-01-22

Accept (Poster)